Systematic revision of Platanthera in the Azorean archipelago: not one but three species, including arguably Europe’s rarest orchid

Bateman Richard M. 1 r.bateman@kew.org
Rudall Paula J. 1
Moura Mónica 2
1 Royal Botanic Gardens Kew, Richmond, Surrey , United Kingdom
2 CIBIO Research Center in Biodiversity and Genetic Resources – Azores, Department of Biology, University of the Azores , Ponta Delgada , Portugal
Roberts David
Electronic publication date: 2013 Dec 10
Publication date: 2013
Volume: 1
Electronic Location ID: e218
Received 2013 Aug 30; Accepted 2013 Nov 8
Copyright: © 2013 Bateman et al.
Copyright year: 2013
Copyright holder: Bateman et al.
License: This is an open access article distributed under the terms of the Creative Commons Attribution License, which permits unrestricted use, distribution, and reproduction in any medium, provided the original author and source are credited.
License URL: https://creativecommons.org/licenses/by/3.0/

Keywords: Endemism, Evolutionary radiation, Migration, Molecular phylogeny, Monography, Morphometrics, Orchid, Platanthera, Species circumscription, Speciation

Funding: Systematics Association Bentham-Moxon Trust Small grants to support fieldwork in the Azores were kindly provided by the Systematics Association (to RMB) and the Bentham-Moxon Trust (to PJR). The funders had no role in study design, data collection and analysis, decision to publish, or preparation of the manuscript.

==============================
Background and Aims. The Macaronesian islands represent an excellent crucible for exploring speciation. This dominantly phenotypic study complements a separate genotypic study, together designed to identify and circumscribe Platanthera species (butterfly-orchids) on the Azores, and to determine their geographic origin(s) and underlying speciation mechanism(s).

Methods. 216 individuals of Platanthera from 30 Azorean localities spanning all nine Azorean islands were measured for 38 morphological characters, supported by light and scanning electron microscopy of selected flowers. They are compared through detailed multivariate and univariate analyses with four widespread continental European relatives in the P. bifolia-chlorantha aggregate, represented by 154 plants from 25 populations, and with the highly misleading original taxonomic descriptions. Physiographic and ecological data were also recorded for each study population.

Key Results. Despite limited genetic divergence, detailed phenotypic survey reveals not one or two but three discrete endemic species of Platanthera that are readily distinguished using several characters, most floral: P. pollostantha (newly named, formerly P. micrantha) occupies the widest range of habitats and altitudes and occurs on all nine islands; P. micrantha (formerly P. azorica) occurs on eight islands but is restricted to small, scattered populations in laurisilva scrub; the true P. azorica appears confined to a single volcanigenic ridge on the central island of São Jorge.

Conclusions. Although hybridity seems low, the excess of phenotypic over genotypic divergence suggests comparatively recent speciation. The most probable of several credible scenarios is that Azorean Platantheras represent a single migration to the archipelago of airborne seed from ancestral population(s) located in southwest Europe rather than North America, originating from within the P. bifolia-chlorantha aggregate. We hypothesise that an initial anagenetic speciation event, aided by the founder effect, was followed by the independent origins of at least one of the two rarer endemic species from within the first-formed endemic species, via a cladogenetic speciation process that involved radical shifts in floral development, considerable phenotypic convergence, and increased mycorrhizal specificity. The recent amalgamation by IUCN of Azorean Platantheras into a single putative species on their Red List urgently requires overruling, as (a) P. azorica is arguably Europe’s rarest bona fide orchid species and (b) the almost equally rare P. micrantha is one of the best indicators of semi-natural laurisilva habitats remaining on the Azores. Both species are threatened by habitat destruction and invasive alien plants. These orchids constitute a model system that illustrates the general advantages of circumscribing species by prioritising field-based over herbarium-based morphological approaches.

Introduction

Given that they are situated 1600 km from the closest continental landmass of Iberia, 2000 km from the Atlas Mountains of Morocco and 2300 km from Newfoundland, it is perhaps not surprising that the nine islands that constitute the Azores archipelago (total area 2335 km2) support an exceptionally impoverished orchid flora. Only two orchid genera are represented on the islands.

Firstly, a unique outlier of the otherwise exclusively Mediterranean tongue-orchids of the genus Serapias has attracted some attention. Assigned to S. cordigera when first reported from five Azorean islands of the central and western groups by Seubert & Hochstetter (1843) and Seubert (1844), the more widespread species was later segregated as an Azorean endemic solely on the basis of morphological differences that are at best subtle. This species was initially named S. azorica (Schlechter, 1923), but nomenclatural complications led to its eventual re-description as S. atlantica, following a thorough morphometric survey by Rückbrodt & Rückbrodt (1994). A subsequent study by Delforge (2003) upheld the endemic status attributed to this species, which has now been recorded on all Azorean islands but the western group of Corvo and Flores (Schäfer, 2002; Silva et al., 2010; Silva, 2013), though it is said to be in decline (Tyteca & Gathoye, 2012). More recently, populations of the widespread Mediterranean autogam S. parviflora have been found on the Azorean islands of Terceira and Santa Maria, occupying a North–South-oriented strip in the western half of the latter island (Silva, 2013). Unfortunately, these Serapias species have not yet attracted molecular research.

Greater scientific attention has been paid to the second of the two Azorean orchid genera, Platanthera. These butterfly-orchids formed part of the first serious botanical collections made on the islands. Gathered by KCF Hochstetter in 1838, they featured in a subsequent floristic list (Seubert & Hochstetter, 1843) and were then rapidly (if poorly) described in Seubert’s (1844) landmark flora of the islands. Seubert described two species, initially assigned (incorrectly) to the genus Habenaria: H. micrantha and H. longebracteata. When these species were correctly re-assigned to Platanthera by Schlechter (1920, 1923), nomenclatural rules required renaming of ‘H.’ longebracteata as P. azorica. The epithets micrantha and azorica have since enjoyed common usage. Unfortunately, re-examination of the original Hochstetter specimens during the present investigation has shown unequivocally that the holotype previously viewed as the basis for the morphological species commonly known as P. micrantha is in fact attributable to the morphological species commonly known as P. azorica. The holotype of the species commonly known as P. azorica is in turn attributable to a new and exceptionally rare species of Platanthera, formally described here for the first time (but previously illustrated by Seubert, 1844). This nomenclatural faux pas is especially unfortunate as it leaves the most widespread species, previously known as P. micrantha, lacking a valid epithet. Following with great reluctance the equally regrettable requirements of the International Code of Nomenclature for Algae, Fungi, and Plants (ICN: McNeill et al., 2012), this well-known, relatively frequent, minute-flowered species is here re-described as P. pollostantha, sp. nov.

Thus, it is essential to understand that, throughout the remainder of this text, the widespread taxon long misidentified (and formerly known) as P. micrantha is correctly named P. pollostantha and the taxon long misidentified (and formerly known) as P. azorica is correctly named P. micrantha. The true P. azorica is exceptionally rare and has long been overlooked by fieldworkers. All three species are undoubtedly endemic to the Azores.

Having finally cut this long-standing nomenclatural ‘Gordian knot’, it is important to note three further points: (1) there has been much debate in the literature regarding whether one species or two species of Platanthera occur on the Azores, (2) the majority of observers expressing opinions on this matter have not actually visited the islands, and (3) no previous author has argued that three species of Platanthera occur on the islands, rather than one or two. Indeed, the recent conservation assessment for IUCN Red Listing (Rankou, Fay & Bilz, 2011a) controversially treated all Azorean Platantheras as a single widespread species, P. ‘micrantha’. With this noteworthy exception, and in the absence of knowledge of the existence of the rarest species, conservation attention has largely focused on the species of intermediate rarity, P. micrantha (formerly P. azorica) – a species that our field investigations have shown to be a valuable indicator of high-quality semi-natural vegetation on the islands.

Setting aside taxonomic controversies, these orchids are also of considerable evolutionary interest. Firstly, the Azorean Platantheras constitute the only orchid lineage present in any of the Macaronesian archipelagos that appears to have undergone dichotomous (cladogenetic) speciation following its arrival on the islands (Bateman et al., in press). Also, all of the few previous authors who have speculated on the likely origin of the continental migrant(s) that are assumed to have established the genus on the Azores have favoured an origin from North America and/or the Palearctic rather than from Iberia or North Africa (e.g., Delforge, 2003).

Recent papers presented DNA evidence from nuclear ribosomal Internal Transcribed Spacer (ITS) sequences that the Azorean butterfly-orchids are correctly assigned to Platanthera rather than Habenaria, clearly placing them within the P. bifolia-chlorantha group that is the sole representative of the genus in southern and central continental Europe (Bateman et al., 2009; Bateman, James & Rudall, 2012). Thus, an origin of the lineage from North America rather than from Europe or North Africa can be confidently rejected. Further molecular data are featured in a companion paper to the present work; comparison by Bateman et al. (in press) of nrITS and plastid microsatellite data from Azorean and European plus North African species of the P. bifolia-chlorantha aggregate suggests (albeit equivocally) that the Azorean lineage represents a single migration of seed to the archipelago from a mainland European population within the aggregate, followed by a modest radiation of the immigrant lineage on the islands into three endemic species. Moreover, ITS data derived from the fungal symbionts of the plants indicate that mycorrhizal specialisation played a role in these speciation events (Bateman et al., in press).

The present paper focuses on the results of a detailed and intensive field-based morphometric survey of populations distributed across the nine islands, considered in the context of both the molecular data and a historical and herbarium-based review. We use this ‘integrated monograph’ to address the following questions:

(1) Can we confirm that not one or two but rather three bona fide biological species of Platanthera occur on the Azores?

(2) Can the nomenclatural puzzles long surrounding their Linnean epithets finally be fully explained, and both satisfactorily and legally resolved?

(3) Do morphological characters exist that are capable of reliably distinguishing among these species and separating them from their continental relatives?

(4) What is the frequency of hybridisation between these species?

(5) What are their habitat preferences and biogeographic distributions?

(6) Which mechanisms are implicated in their respective speciation events?

(7) Can we further clarify their relationships?

(8) What are the major threats to, and most appropriate conservation status for, each re-circumscribed species?

We also compare the broader implications of this study for pursuing field-based versus herbarium-based approaches to taxonomic revision.

Materials and Methods

Field sampling

In total, seven species of Platanthera were sampled: three from the Azores (Figs. 1–6) and four from mainland Eurasia (Figs. 7–10). Of the 21 colonies of Platanthera in southern England visited by RB and PR during May–June 2003 and/or 2004 that contained flowering plants capable of yielding useful data, 11 contained P. chlorantha only (Fig. 10), eight contained P. bifolia only (Fig. 9), and two contained both species. Small numbers of putative hybrid plants were also found in both of the mixed colonies (Bateman, James & Rudall, 2012). For the smaller colonies, all flowering plants in suitable condition were measured (six sites yielded only one measurable plant). In larger colonies, individual plants were selected to adequately represent the range of phenotypic variation and habitat occupancy evident at the locality. In total, measurements were taken from 139 plants: 79 individuals of P. chlorantha, 55 individuals of P. bifolia and five putative hybrids between these species. Information describing the 21 sampled localities was summarised in appendices 1 and 2 of Bateman, James & Rudall (2012).

Figure 1 Azorean Platanthera populations sampled for morphometric analyses during the present study.

(A) shows the relative geographic positions of the three island groups, (B) western group, (C) central group, (D) eastern group. Populations shown in red are P. pollostantha only, populations in green are P. micrantha only, populations in yellow mix P. pollostantha and P. micrantha, populations in blue mix P. pollostantha and P. azorica. Base maps: (A) from Fig. 1B of Schäfer et al. (2011), (B–D) base images courtesy of GoogleEarth. Scale bar = 25 km.

Figure 2 Classic habitats of the Azorean Platanthera species.

(A) Brejos Plateau, Pico (ca 790 m: P. pollostantha, P. micrantha). (B) Caldera, Faial (ca 890 m, crater bottom 570 m: P. pollostantha, P. micrantha). Images: P Rudall.

Figure 3 Habitats of the Azorean Platanthera species (continued).

(A) Alpine grassland panorama looking eastwards from Pico da Esperança, São Jorge (ca 1080 m: P. pollostantha, P. micrantha, P. azorica). (B) Alpine heathland near Capitão, Pico (ca 1000 m: P. pollostantha). (C) Alpine phenotype of P. pollostantha growing at the locality shown in (B). (D) Typical mid-altitude laurisilva habitat supporting both P. pollostantha and P. micrantha at Cabeço da Rocha, Pico. (E) Highly invasive Hedychium gardnerianum choking P. pollostantha at Pico Alto, Santa Maria. Images: (A) = R Poot, (D) = P Rudall, remainder = R Bateman.

Figure 4 Plants and flowers of P. pollostantha.

(A, B) Entire plant and inflorescence. (C, D) Perpendicular and lateral views of flowers. (E) Old (left) and new (right) tubers. (A, B) from Bica, Pico, (C–E) from Lagoa do Canario, São Miguel. Scale bar for (C, D) = 5 mm. Images: R Bateman.

Figure 5 Plants and flowers of P. micrantha.

(A, B) Entire plant and inflorescence. (C, D) Perpendicular and lateral views of flowers. (A, D) from Brejos, Pico (Fig. 2A), (B, C) from Pico Pinheiro, São Jorge. Scale bar for (C, D) = 5 mm. Images: R Bateman.

Figure 6 Plants and flowers of P. azorica.

(A, B) Entire plant and inflorescence. (C, D) Perpendicular and lateral views of flowers. (All from Pico da Esperança, São Jorge (Fig. 3A). Scale bar for (C, D) = 5 mm. Images: R Bateman.

Figure 7 Plants, flowers and habitat of P. holmboei from Mt Olympus, Lesvos.

(A) Habitat – moist, high-altitude chestnut forests. (B, C) Entire plant and inflorescence. (D, E) Perpendicular and lateral views of flowers within two inflorescences. Scale bar for (D, E) = 10 mm (i.e., 20% linear smaller scale than that used in the equivalent close-up images in Figs. 4–6). Images: R Bateman.

Figure 8 Plants, flowers and habitat of P. algeriensis from Ifrane, Morocco (A–E) and Ghisonaccia, Corsica (F).

(A) Habitat – wet flushes surrounding a stream in a semi-arid high-altitude hinterland. (B, C) Entire plant and inflorescence. (D, E) Perpendicular and lateral views of Moroccan flower. (F) Perpendicular view of Corsican flower. Scale bar for (D–F) = 10 mm (i.e., 20% linear smaller scale than that used in the equivalent close-up images in Figs. 4–6). Images: (F) = R Bateman, remainder = R Poot.

Figure 9 Plants and flowers of P. bifolia from the British Isles.

(A, B) Entire plant and perpendicular view of flower, chalk downland and woodland ecotypes, respectively. (C) Lateral and perpendicular views of flower, moorland ecotype. (D) Lateral view of flowers, chalk downland ecotype. (A) from Pewsey Downs, Wiltshire, (B) from Bix Bottom, Oxfordshire, (C) from Broadford, Skye, (D) from Morgan’s Hill, Wiltshire. Scale bar for (B, C) = 10 mm (i.e., 20% linear smaller scale than that used in the equivalent close-up images in Figs. 4–6). Images: R Bateman.

Figure 10 Plants and flowers of P. chlorantha from the British Isles.

(A) Entire plant. (B) Perpendicular view of flower. (C) Partial inflorescence of green-flowered mutant. (D) Partial inflorescence. (A) from Aston Clinton Ragpits, Buckinghamshire, (B) from Bix Bottom, Oxfordshire, (C) from Keltney Burn, Perthshire, (D) from East Hoathly, Sussex. Scale bar for (B) = 10 mm (i.e., 20% linear smaller scale than that used in the equivalent close-up images in Figs. 4–6). Images: (D) Derek Turner Ettlinger, rest R Bateman.

Attempts by RB and PR to extract data from populations of P. algeriensis were only partially successful. In April 2011 we were able to locate only a single flowering plant at an east coast locality on the central Mediterranean island of Corsica (Fig. 8F); fortunately, this one plant eventually generated morphometric and DNA data. Our May 2012 expedition to the Ifrane region of Morocco yielded much larger numbers of individuals of P. algeriensis (Figs. 8A–8E) but as the late season meant that none was close to flowering, we were restricted to obtaining DNA data, albeit from both the orchids and their mycorrhizal symbionts (Bateman et al., in press). The two subpopulations sampled were separated by ca 400 m. In May 2013 we focused on collecting data on P. holmboei from its westernmost occurrence, on the Aegean island of Lesvos. We eventually measured 14 plants from three populations that together constituted a 4 km North–South transect across Mt Olympus (Fig. 7).

All three authors participated in field sampling of Platanthera on the Azores (Table 1), which was conducted under permit by Moura in 2009 and 2011, and by Bateman and Rudall in 2011 and 2012. A total of 30 sites were sampled morphometrically, and a further seven sites yielded DNA samples only. Platanthera pollostantha was sampled on all islands except Graciosa, and P. micrantha on all islands except Graciosa, Terceira and Santa Maria (Fig. 1). Topographic features sampled on these ubiquitously volcanigenic landscapes were most commonly calderas (Fig. 2B), parasitic cones and lava flows, particularly lava tunnels/caves (Fig. 2A). The most common habitats were grassy clearings and tracksides within laurisilva scrub (Figs. 2A and 3D), but also included alpine heaths (Fig. 3B) and tracksides through Cryptomeria plantations (see also illustrations in Tyteca & Gathoye, 2012). Current evidence suggests that P. azorica is confined to a small upland area of São Jorge (Fig. 3A: but see ‘Convoluted taxonomic history’ below). Sampling was especially intensive on Pico, a large, topographically diverse and comparatively unspoilt island located within the central cluster of five islands (Figs. 1, 2A, 3B and 3D).

Table 1 Details of localities sampled for Azorean Platantheras.

								Morphometrics/
DNA			Mycorrhizal
DNA			
Group	Island	Locality	Habitat(s)	pH	Altitude
(m asl)	Peak flowering	Collector(s)	Po	Mi	Az	Po	Mi	Az	
W	Flores	Caldeira Seca e Caldeira
Comprida	V	–	594	–	M11	10	2	–	–	1	–	
		Ribeira da Badanelia	V	5.9	683	–	M11	1	–	–	1	–	–	
	Corvo	Caldeirão (Cumeeiras)	A	4.6	600	–	M11	10	–	–	2	–	–	
		S slope caldera (Morro dos
Homens)	A?	–	551	–	C08	(5)	–	–	–	–	–	
		Forked ribeira, NE caldera
(Cancela)	A?	–	414	–	C08	(1)	(1)	–	–	–	–	
		Lower rd, E coast (Lomba)	V?	–	321	–	C08	–	(6)	–	–	–	–	
C	Faial	Cabeço dos Trinta,
W Caldera	V	–	731	6/3	B11	–	1d	–	–	1	–	
		Canto dos Banquinhos,
Faial caldera	V	–	780	6/4 (po)	B11	3	8	–	2	1	–	
						6/3 (mi)								
	Pico	Furna do Frei Matias	V	4.1	660	–	M09	10	1	–	–	–	–	
		Roadside bank N Capitão,
W Pico Mt	A	–	1000	6/4	B11	6	–	–			–	
		T-junction on EN3,
Cerrado de Sonicas,
NNW Pico Mt	H	–	870	6/2-3	B11	5	–	–	1	–	–	
		Refugium, Cabeço das
Cabras, W Pico Mt	H	–	1235	6/3-4	B12	1	–	–	1	–	–	
		Lava tunnel, roadside EN3
imm. E Bica, N Pico Mt	H + V	–	928	6/2-3	B12	3	2	–	2	2	–	
		Chã do Pelado, Caveiro	L	5.3	790	–	M09	10	–	–		–	–	
		Pico da Urze rd S EN3,
S Cabeço de Teixo,
SW São Roque	V	–	830	6/2	B12	3	5	–	3	5	–	
		Brejos plateau, N Cabeço
do Sintrão, ENE Pico Mt	V	–	789	6/4 (po)	B11	7	7	–	4	5	–	
						6/3-4 (mi)	B12							
		Track on plateau
W Cabeço da Rocha,
NNW Lajes	V + H	–	774	6/3-4 (po)	B11	2	4	–	2	1	–	
						6/2-3 (mi)								
		Track E imm. S junction
EN3-EN2, Corre Agua,
S São Roque	H	–	730	6/2-3	B12	(1)	–	–	–	–	–	
		Roadside NW Cabeço do
Ferrobo, NW Lajes	V	–	540	6/2	B12	(1)	–	–	–	–	–	
		Imm. E Cabeco de Rocha,
W Lagoa do Caiado, N Lajes	L	–	840	6/2	B12	(1)	–	–	–	–	–	
		Track, Achada,
N Lagoa do Paul, SE Grotões, N Lajes	V	–	830	6/2	B12	(1)	–	–	–	–	–	
		Caldeirão da Ribeirinha	V	–	750	–	M09	10	1	–	–	–	–	
	São Jorge	Small volcanic cone, W Pico da Esperança	A + V	5.9	976	–	M11	3	–	8	–	–	2	
		Trackside,
SE slope Pico da Esperan ç a,
SSW Norte Grande	A	–	1000	6/4 (po)	B12	6	–	12	2	–	8	
						6/1-2 (az)								
		Ravine N Pico do Areeiro, S Norte Grande	A + V	–	885	6/2-3	B12	+	1	–	–	–	–	
		Crater, Pico Pinheiro-P. do Areeiro,
S Norte Grande	A + V	–	860	6/4 (po)	B12	2	6	–	2	4	–	
						6/2 (mi)								
		Track NE Pico Pinheiro,
S Norte Grande	V	–	730	6/2-3	B12	+	1	–	–	–	–	
		Woodland E Pico do Paul,
N Calheta	L	–	468	6/1-2	B12	2	–	–	1	–	–	
		Trilho Topo-Fajã do Santo Cristo	V	5.2	560	–	M11	10	6a	–	2	1	–	
	Graciosa	[NOT SAMPLED]												
	Terceira	Rocha do Chambre	V	–	549	–	M11	6	–	–	2	–	–	
		Caminho Algar do Carvão-Caldeira de Agualva	V	–	610	–	M11	2	–	–	–	–	–	
E	São Miguel	Portal de Vento, E Vista
do Rei, Sete Cidades	H + G	–	610	6/3-4	B11	5	–	–	5	–	–	
		E + N Lagoa do Canário,
SE Sete Cidades caldera	C	–	810	6/3	B11	5	–	–	4	–	–	
		Lagoa do Fogo, SSE Ribeira Grande	V	–	670	6/3-4	B11	–	10	–	–	3	–	
	Sta. Maria	Pico Alto, W Santa Bárbara	V	5.7	474	–	M09	10d	–	–	–	–	–	
		Halfway, rd Pico Alto-Cruz
dos Picos, W Santa Barbara	V	–	468	5/4-6/1	B12	5	–	–	3	–	–	
		Miradouro da Pedra Rija	H + C	6.1	393	–	M09	4b	–	–	–	–	–	
n								141	55 a	20	39	24	10	
Notes.

a Two plants were subsequently tentatively identified as the hybrid P. pollostantha × micrantha.

b One additional DNA sample collected.

c Two additional DNA samples collected.

d Three additional DNA samples collected. Parenthetic figures lack morphometric data and were collected only as DNA samples.

+ species present but not sampled

Taxa

Po P. pollostantha

Mi P. micrantha

Az P. azorica

Collectors

M Mónica Moura

B Richard Bateman and Paula Rudall

C Mark Carine; suffixed numbers indicate the year of collection

Habitat

V laurisilva including Vaccinium cylindraceum

L laurisilva lacking Vaccinium

H Erica heath

A alpine grassland

C Cryptomeria plantation

For peak flowering estimates, the slash is preceded by the month and succeeded by the week(s) of that month.

In total, morphometric data were obtained from 141 plants of P. pollostantha (26 localities – Fig. 4), 55 plants of P. micrantha (two subsequently re-assessed as hybrids: 13 localities – Fig. 5), and 20 plants of P. azorica (two adjacent localities, where this species co-occurred with P. pollostantha – Fig. 6), totalling 216 plants. Numbers of samples available for DNA analyses were slightly larger (Table 1). Subsets of the individuals measured were sampled for DNA analysis of mycorrhizal associates: 39 plants of P. pollostantha (17 localities), 24 plants of P. micrantha (two subsequently re-assessed as hybrids: 10 localities), and 10 plants of P. azorica (two localities) (Bateman et al., in press). Pickled flowers for microscopic study and imaging were obtained from several populations spanning the three groups of islands.

Our within-site sampling strategy was designed to minimise disturbance to individual plants. Destructive measurements of tubers were not attempted, and the two or three stem-roots present were disturbed only in a minority of plants that were subjected to not only morphometric study but also sampling for mycorrhizae. Within each population, plants for study were chosen to proportionately reflect the range of variation evident in both morphology and habitat. Vegetative characters were measured non-destructively from in situ plants, and only approximately five flowers from each plant were removed for further study: one was permanently mounted and measured, whereas the remainder were placed in fine-grained dried silica gel to act as a DNA-friendly voucher. Wherever possible, the florets chosen to provide morphometric data on the flower, ovary and bract were located 30–40% of the distance from the base to the apex of the inflorescence, in order to minimise the widespread effect of diminution in flower size toward the apex (Bateman & Rudall, 2006).

Morphometric characters

Largely following Bateman, James & Rudall (2012), the 38 characters that were scored morphometrically (Appendix 1) described the stem and inflorescence (4), leaves (7), bracts (5), labellum (5), spur and ovary (5), sepals and lateral petals (5), and gynostemium (7). They can alternatively be categorised as metric (27), meristic (3), multistate-scalar (6), and operationally bistate (2). Metric characters for most floral organs were measured at a resolution of 0.1 mm; RB and PR used a Leitz ×8 graduated ocular, whereas MM used electronic calipers. There were two exceptions: RB measured gynostemium characters for some individuals to a resolution of 0.1 mm at ×10 magnification under a Leica MZ8 binocular microscope, and recorded floral bract cells (two characters) in µm at ×100 magnification under a Leica Dialux 20 compound microscope. The complete absence of anthocyanin pigments from the clade rendered redundant our usual practice of quantitatively colour matching various flower-parts.

Data analysis

Morphometric data for individual plants were summarised on an Excel v14.3 spreadsheet. Mean values, plus sample standard deviations and coefficients of variation for all metric and some meristic characters, were calculated for every character in each of the three species. Univariate and bivariate analyses were summarised and presented using Deltagraph v5.6 (SPSS/Red Rock software, 2005), which in some cases was also used to calculate linear regressions.

The full morphometric matrix contained 370 individuals ×38 characters. That part of the matrix consisting of the 139 plants of P. bifolia and P. chlorantha inherited from the study of Bateman, James & Rudall (2012), plus the 15 plants of P. algeriensis and P. holmboei measured subsequently, contained 13.0% missing values, whereas the 216 plants measured in the Azores incurred only 4.6% missing values. The characters affected by missing values on the Azores were auricle length (C21), bract cell diameter (C22) and shape (C23), basal bract length (C26) and position of maximum leaf width (C35); of these, only the bract cell characters incurred more than one-third of missing values. The assembled data were analysed by multivariate methods using Genstat v14 (Payne et al., 2011). All calculated ratios were also omitted from the multivariate analyses as, by definition, they duplicated their constituent characters.

One character (C4: pale green versus dark green pigmentation of the labellum) was subsequently judged to largely duplicate another character (C5: maximum extent of green pigmentation on the labellum) and was therefore omitted from the analyses. The remaining 37 characters were used to compute a symmetrical matrix that quantified the similarities of pairs of data sets (i.e., plants) using the Gower Similarity Coefficient (Gower, 1971) on unweighted data sets scaled to unit variance. The resulting matrix was in turn used to construct a minimum spanning tree (Gower & Ross, 1969) and subsequently to calculate principal coordinates (Gower, 1966; Gower, 1985) – compound vectors that incorporate positively or negatively correlated characters that are most variable and therefore potentially diagnostic. Principal coordinates are especially effective for simultaneously analysing heterogeneous suites of morphological characters and can comfortably accommodate missing values; they have proven invaluable for assessing relationships among orchid species and populations throughout the last three decades (reviewed by Bateman, 2001).

Six separate multivariate analyses were conducted, all but one involving the progressive reduction in the number of taxa (and thus of plants) included: all seven species, the three Azorean species only, the three Azorean species only (vegetative characters omitted), the two more widespread (and widely accepted) Azorean species only, and each of these two species alone (these single-species analyses were designed primarily to investigate relationships between populations on different islands). For each multivariate analysis, the first four principal coordinates (PC1–4) were plotted together in pairwise combinations to assess the degree of morphological separation of individuals (and thereby of populations and taxa) in these dimensions, and pseudo-F statistics were obtained to indicate the relative contributions to each coordinate of the original variables.

Micro-imaging

Selected flowers of the two British species were sampled from the Stockbury area of north-central Kent and stored in 70% ethanol. The spirit collection at RBG Kew yielded an alcohol-fixed inflorescence of P. holmboei from Mt Troodos on Cyprus, later supplemented with flowers obtained in 2013 from several plants on Mt Olympus, Lesvos. Flowers of P. algeriensis were collected in April 2011 by RB and PR from a single plant located along the east coast of Corsica near Ghisonaccia. Flowers of several accessions of all three Azorean taxa were placed in alcohol by MM in June/July 2009, and by RB and PR in June 2011 (several localities on Pico) and June 2012 (all from the ‘spinal ridge’ linking Pico da Esperança to Pico Areeiro).

Specimens were initially imaged using a Nikon Shuttlepix P-MFSC optical system, where necessary subsequently using EDX image stacking to achieve an average focus from multiple primary optical frames. Preparation for scanning electron microscopy (SEM) involved selecting flowers from each inflorescence for dehydration through an alcohol series to 100% ethanol. They were then stabilised using an Autosamdri 815B critical-point drier, mounted onto stubs using double-sided adhesive tape, coated with platinum using an Emtech K550X sputter-coater, and examined under a Hitachi cold-field emission SEM S-4700-II at 2 kV or 4 kV. The resulting images were recorded digitally for subsequent manipulation in Adobe Photoshop. Comparison of fresh and spirit material of P. bifolia demonstrated the absence of any serious artefacts caused by preservation in spirit.

Journal nomenclatural statement

The electronic version of this article in Portable Document Format (PDF) will represent a published work according to the International Code of Nomenclature for algae, fungi, and plants (ICN), and hence the new names contained in the electronic version are effectively published under that Code from the electronic edition alone. In addition, new names contained in this work which have been issued with identifiers by IPNI (International Plant Names Index) will eventually be made available to the Global Names Index. The IPNI LSIDs can be resolved and the associated information viewed through any standard web browser by appending the Life Science Identifier (LSID) contained in this publication to the prefix “http://ipni.org/”. The online version of this work is archived and available from the following digital repositories: PeerJ, PubMed Central, and CLOCKSS.

Results

Micro-imaging

Both light and scanning electron micro-imaging were performed primarily to detail more accurately the morphology of the gynostemium, though this approach also provided useful data on the epidermal micromorphology of the perianth segments and spur interior. The resulting images (Figs. 11–17) support some generalisations previously made regarding the floral morphology of the P. bifolia-chlorantha clade but also provide some valuable new insights. The gynostemia of these species are characterised by pronounced connectives, large tripartite stigmas, ‘granular’ (perhaps better described as botryoidal) auricles, and well-developed anther locules containing tripartite pollinaria. The pollinia are sectile, consisting of two longitudinal rows of massulae linked by elastoviscin threads, and the viscidia protrude to varying degrees into the pre-stigmatic cavity. The sepals and lateral petals reliably produce stomata adaxially, but in contrast, the spur interiors differ considerably in epidermal features.

Figure 11 Light micrographs of Platanthera flowers.

(A) Flower, P. pollostantha. (B) Flower, P. micrantha. (C) Flower, P. azorica. (D) Flower, P. algeriensis (Corsica). Additional images show (E) pollen massulae attached to the three stigma lobes of P. pollostantha, and (F) the compact, partially disaggregated pollinium and partially collapsed viscidium of P. micrantha. The labellum and lateral sepals have been removed to expose the gynostemium of each flower. Images: P Rudall. Scales = 1 mm (A–E), 0.5 mm (F).

Figure 12 Scanning electron micrographs of flowers of British P. bifolia.

(A) Flower with median sepal removed, showing partially obscured circular spur entrance. (B) Oblique view of gynostemium. (C) Details of discoid viscidium, stigma and auricles. (D) Stomata scattered across the adaxial surface of the median sepal. (E) Strongly papillate cells lining the interior of the labellar spur. Images: P Rudall.

Figure 13 Scanning electron micrographs of flowers of P. pollostantha.

(A) Intact flower. (B) Oblique view of gynostemium featuring anther locules and pollinaria. (C) Flower with both pollinaria removed and massulae deposited on the three stigma lobes. (D) Stomata present on the adaxial surface of the median sepal. (E) Putatively glandular cells on the adaxial surface of the labellum towards its apex. (F) Smooth cells lining the interior of the labellar spur. Images: P Rudall.

Figure 14 Scanning electron micrographs of flowers of P. micrantha.

(A) Intact flower. (B) Perpendicular view of gynostemium of pre-anthetic flower, showing thickened margin of the anther locules and hydrated viscidial discs. (C) Gynostemium of mature flower, featuring anther locules, pollinaria and ‘letter box’ spur entrance. (D) Flower nearing senescence, with both pollinaria largely disaggregated. (E) Smooth cells lining the interior of the labellar spur. Images: P Rudall.

Figure 15 Scanning electron micrographs of flowers of P. azorica.

(A) Flower with labellum removed, showing circular spur entrance. (B) Perpendicular view of gynostemium showing one anther locule containing a pollinarium, featuring the geniculate caudicle and discoid viscidium. (C) Putatively glandular cells on the adaxial surface of the labellum towards its apex. (D) Strongly papillate cells lining the interior of the labellar spur and bearing nectar residues. Images: P Rudall.

Figure 16 Scanning electron micrographs of flowers of Corsican P. algeriensis.

(A) Flower with labellum removed, showing circular spur entrance. (B) Perpendicular view of gynostemium showing one anther locule containing a pollinarium, featuring the geniculate caudicle and discoid viscidium. (C) Dissected spur, showing the preferential development of papillae above the two arms of the vascular bundle that loops dorsiventrally through the spur. (D) Strongly papillate cells lining the interior of the labellar spur, bearing nectar residues. Images: P Rudall.

Figure 17 Scanning electron micrographs of flowers of P. holmboei from Cyprus (C) and Lesvos (remainder).

(A) Slightly oblique view of flower lacking both pollinaria. (B) Near-perpendicular view of gynostemium, featuring auricles, circular spur entrance, and tripartite stigma bearing pollen massulae. (C) Details of proximal portion of pollnarium and viscidium from bud approaching anthesis. (D) Details of proximal portion of the locule showing the distinctive recess previously occupied by the viscidium. (E) Dissected spur, showing the preferential development of papillae above the two arms of the vascular bundle that loops dorsiventrally through the spur. (F) Strongly papillate cells lining the interior of the labellar spur, bearing nectar residues. Images: P Rudall.

Perianth segments

The flower of P. pollostantha (LSID: 77134154-1) depicted in Fig. 13A is a relatively recently opened bud – the labellum has not yet reached a near-vertical position or become recurved – and the lateral sepals have been removed to reveal the compact (and somewhat disrupted) gynostemium and the dorsiventrally compressed entrance to the remarkably short labellar spur. The median sepal combines with the lateral petals to form a tight hood cowling the gynostemium. The sepals and lateral petals bear stomata adaxially (Fig. 13D), whereas the labellum shows some evidence of glandular cells concentrated towards the apex (Fig. 13E). The internal epidermis of the spur is smooth (Fig. 13F).

The flower of P. micrantha depicted in Fig. 14A remains intact, though the labellum has deliberately been torn at the base and forced downward in order to better expose the gynostemium. The gynostemium and dorsiventrally compressed spur entrance are more effectively detailed in Figs. 14B and 14C. The spur curves strongly forward, projecting toward the viewer from beneath the upwardly-curved labellum (Fig. 14A). The median sepal forms a more-or-less planar ‘awning’ above the gynostemium, which is also protected by the two lateral petals that twist inwards to form a distinctive arch above the gynostemium, their apices sometime overlapping. In contrast, the lateral sepals are spreading and oriented closer to the vertical than the horizontal. The interior of the spur is smooth (Fig. 14E).

The gynostemium of P. azorica is again protected by a hood that consists of the lateral petals and median sepal, but both the gynostemium and hood are more elongate (Fig. 15A). The stigma is well developed and its lateral lobes extend downward on either side of the circular spur entrance, overhung by a substantial rostellum. Once again, the sepals and petals bear stomata, the labellum appears distally glandular (Fig. 15C), but in this species, the interior of the spur is strongly papillate rather than smooth (Fig. 15D).

Flowers of three of the four large-flowered mainland species are remarkably similar structurally, resembling P. azorica and showing only modest differences in flower size and proportions of particular organs. The flower of P. algeriensis (Fig. 16A) is remarkably similar to that of P. azorica, differing mainly in size, while that of P. algeriensis (Fig. 17A) in turn closely resembles the flower of P. chlorantha (multiple SEM images illustrated in Fig. 4 of Bateman, James & Rudall, 2012). All three species have stomata on sepals and petals, an apparently glandular labellum, and a strongly papillate spur interior (Fig. 14D). Moreover, the distal, nectar-secreting portions of the spurs of the mainland species are oval in transverse section, being expanded dorsiventrally toward the single vascular strand that loops around the spur apex (cf. Box et al., 2008; Bell et al., 2009). In these species, the papillae are best developed in two longitudinal zones located immediately above the opposing arms of the vein (Figs. 16C and 17E).

These perianth features also characterise P. bifolia (Figs. 12D and 12E), which differs from P. chlorantha, P. algeriensis, P. holmboei and P. azorica mainly in gynostemial structures (see below). In at least some individuals, two pairs of gynostemial projections partially obscure the cylindrical spur entrance, the inner pair extending from beneath the anther locules (interpreted as extensions of the lateral lobes of the stigma) and the outer flanges curving inwards from below the auricles (Fig. 12A). Admittedly, these structures are less well-developed in some other individuals of this species (Fig. 12B). As in the other large-flowered species, sepals and labellum bear stomata (Fig. 12D), and the interior of the spur is strongly papillate (Fig. 12E).

Gynostemium

The wide range of gynostemium morphologies exhibited by the genus Platanthera s.l. was surveyed by Efimov (2011), and some details of P. bifolia, P. chlorantha and P. holmboei were illustrated by Claessens & Kleynen (2011) and Bateman, James & Rudall (2012). The gynostemia of the three Azorean species (Figs. 11A–11C, 11E, 11F and 13–15) share the basic architecture that is characteristic of the bifolia-chlorantha aggregate (Figs. 11D, 12, 16 and 17) – an upright orientation; gynostemium flanked by two pale, granular auricles; prominent paired anther locules linked by a robust connective; a slightly concave tripartite stigma, the larger polygonal median lobe being flanked by two triangular lateral lobes (all typically coated in a pale, viscous stigmatic fluid); a laterally extended but often subdued rostellar ledge located immediately above the stigma; viscidia exposed rather than being enclosed in a bursicle; pollinaria readily divisible into sectile pollinium and caudicle that are undoubtedly anther-derived versus a circular/oval, concave, at least obscurely bipartite viscidium that is reputedly stigma-derived (cf. Kurzweil, 1987; Claessens & Kleynen, 2011).

The plates detailing the gynostemia of our study species (Figs. 11–17) clearly separate P. micrantha plus P. pollostantha from the remaining five species. Moreover, P. micrantha and P. pollostantha resemble P. bifolia more closely than P. azorica or P. chlorantha and its relatives (P. algeriensis and P. holmboei). Platanthera azorica and the P. chlorantha group have circular spur entrances, circular viscidia, geniculate and terete caudicles, pollinia consisting of several vertical rows of massulae, sigmoid locular apertures, well-developed auricles and, most importantly, large, collar-like stigmatic surfaces extensively coated in milk-coloured stigmatic fluid and delineated above by a laterally extended rostellum (Figs. 11C, 11D and 15–17). In contrast, P. micrantha and P. pollostantha have dorsiventrally compressed ‘letter-box’ spur entrances, oval viscidia, more-or-less linear strap-like caudicles, fewer vertical rows of massulae, linear locular apertures, auricles that are often barely discernible, and small, dorsiventrally compressed stigmatic surfaces where often only the central lobe is coated with stigmatic fluid and the rostellum is both short and subdued (Figs. 11A, 11B, 11E, 11F, 12 and 13).

Remarkably, no structural or even proportional differences distinguish the Azorean P. azorica from the mainland P. algeriensis; these species differ primarily in the somewhat smaller flower size of the former. The apically broader connective of P. holmboei (Figs. 17A and 17B) causes it to more closely resemble P. chlorantha, though again its flowers tend to be somewhat smaller. The similarity among these four species is particularly striking in their shared possession of a distinctive pollinarium morphology. Their long caudicles undergo a right-angled bend just above the viscidia, thereby positioning the pair of circular, sucker-like viscidia in opposition, so that they are well-placed to contact the compound eyes of a suitably sized insect visitor (cf. Figs. 11C, 11D and 15–17).

Strong similarities are also evident between the gynostemia of the two small-flowered Azorean species – P. pollostantha and the somewhat larger-flowered P. micrantha (cf. Fig. 11A, 13 vs Fig. 11B, 14) – though some subtle differences are discernible. The stigma of P. pollostantha has a larger height-to-width ratio, and lateral lobes that project outward as rather subdued ‘horns’. Both species have distinctive pollinaria with strap-like caudicles that, despite their comparatively short length, project below the rostellar ledge into the void beneath, each being located immediately in front of one of the lateral lobes of the stigma and diagonally above the spur entrance. The viscidia project downward but also tend to be angled slightly backward, seemingly well-placed to contact any insect proboscis that is actively seeking the spur entrance. This posture also characterises at least some varieties of P. bifolia (Fig. 12; see also p. 283 of Claessens & Kleynen, 2011), and is already evident in immature buds (Fig. 14B). However, the viscidia of the two Azorean species differ in detail. Those of P. pollostantha are near-circular in outline and their reputed bipartite nature is obscure (Fig. 13B), whereas the bipartite nature of the viscidia is clear in P. micrantha; the caudicle terminates in a robust, circular inner disc resembling that of P. pollostantha, but this is attached to the centre of a much more extensive, elongate-oval disc. This outer disc consists of less robust tissue that curls downward laterally, thereby forming a hemi-cylinder that is oriented towards the stigmatic surface (Figs. 11F and 14B–14D); it appears to become desiccated soon after anthesis (cf. Figs. 14B versus 14A, 14C and 14D).

Interestingly, relatively large, elongate viscidia also characterise at least some populations of P. bifolia (e.g., p. 284 of Claessens & Kleynen, 2011), though in other populations of this species the viscidia are directed inward (Fig. 12B) and resemble more closely those of P. chlorantha and its Mediterranean endemic allies (Figs. 16 and 17).

Multivariate analyses

Seven taxa

The principal coordinates analysis of 370 individuals for 37 characters gave reliable separation of five of the seven species included, but the plot of PC1 versus PC2 allowed slight overlap between P. azorica and P. holmboei and substantial overlap between P. pollostantha and P. micrantha (Fig. 18A). The first axis accounted for a remarkably high percentage of the total variance and separated the species into four clusters: pollostantha plus micrantha, azorica plus holmboei plus bifolia, algeriensis (single plant only analysed), and chlorantha, on the basis of broadly positively correlated gradation in the sizes of all flower parts, most notably labellum and spur lengths, plus lateral petal colour (Table 2A). The second (and much weaker) axis reliably separated azorica plus holmboei from bifolia and, with less success, pollostantha from micrantha – this axis was influenced by ‘vigour’ characters sensu Bateman & Denholm (1989) such as leaf, bract and inflorescence dimensions. The third axis served no taxonomic function, being dictated by the angle subtended by the basal leaves relative to the soil surface, while the even weaker fourth axis used an admixture of largely non-diagnostic characters to partially separate azorica from pollostantha and bifolia. Overall, the plot of PC1 versus PC2 closely resembled the result obtained by Bateman, James & Rudall (2012, their Fig. 5) when analysing corresponding data for the Eurasian mainland species P. chlorantha and P. bifolia only.

Figure 18 Principal coordinates plots for the first two axes for two different combinations of taxa and characters (parenthetic figure indicate the percentage of the total variance accounted for by each axis).

(A) Seven taxa, all characters (holotypes excluded). (B) Three taxa, floral characters only. For characters contributing to each axis see Tables 2A and 2C.

Table 2 Variables contributing to the first four principal coordinates of each of six multivariate analyses (cf. Figs. 18 and 19), listed in order of decreasing contribution.

(A) Seven taxa, all 37 characters. (B) Three taxa, all 37 characters. (C) Three taxa, 21 floral characters only. (D) Two taxa, all 37 characters. (E) Platanthera pollostatha only, all 37 characters. (F) Platanthera micrantha only, all 37 characters. Numbers of variables match character numbers given in Appendix 1. Character 4 was omitted from all analyses, and characters 22–37 were also omitted from analysis C. Characters 14A and 23 were invariant in analyses B–F. Double slashes separate dominant from subdominant characters. Italicised characters increase in value toward the positive end of the coordinate, whereas roman characters increase in value towards the negative end of the coordinate.

Principal
coordinate	Percentage of variance
accounted for	Contributing characters	
(A) 370 plants			
1	62.7	1, 6, 12, 18, 14, 15 // 13, 16, 20, 17, 14A, 8, 21, 19, 2 // 10, 7, 36, 23	
2	10.9	24, 33, 25, 30 // 28, 29, 34, 27, 9, 14A	
3	5.6	37 // 27, 34	
4	4.2	32 // 9, 29, 5, 33, 28, 30, 7	
(B) 219 plants			
1	38.6	1, 12, 14 // 13, 17, 10, 25, 6, 18, 16, 15, 19, 20, 24	
2	13.5	29, 27, 33, 28, 3, 32 // 30, 11, 34, 19	
3	8.8	2, 11, 26, 3 // 34, 37, 31, 27, 6	
4	7.3	37, 9, 31, 35, 2, 5	
(C) 219 plants			
1	51.7	1, 12 // 14, 17, 16, 19, 15, 18, 20, 13, 6, 10	
2	14.8	11, 3, 2 // 6	
3	10.4	9	
4	7.3	5	
(D) 198 plants			
1	35.9	1, 12, 14, 24, 10, 25, 33, 13, 6 // 30, 29, 17, 18, 28, 11, 20, 27	
2	11.4	2, 3 // 31, 11	
3	8.9	34, 26 // 28, 27, 5, 37	
4	6.1	9 // 7, 37	
(E) 142 plants			
1	32.3	25, 1, 24, 12, 29, 3014, 13, 2 // 33, 10, 32, 31, 27, 8	
2	13.1	34, 26, 28, 5, 27	
3	7.8	37	
4	6.4	7	
(F) 54 plants			
1	26.3	9, 30 // 12, 1, 32, 10, 13, 33, 16, 25, 28, 29, 2	
2	18.1	11, 26	
3	8.1	5 // 18, 2	
4	7.3	35, 11, 32, 21, 14	

Three taxa (two analyses)

Deleting the four Eurasian species from the matrix and leaving only the Azorean taxa reduced the number of individuals analysed to 219 and rendered one character invariant (C14A: colour of lateral petals). The strength of the first axis decreased relative to that of the second axis (Fig. 19A). The two axes operated together to distribute conspecific individuals diagonally across the plot, suggesting some underlying similarities between the axes. There is strong separation of azorica from the two remaining species, based primarily on the lengths of labellum, lateral petals and lateral sepals, supported by several other flower and bract dimensions (Table 2B). The second axis again largely reflects vegetative vigour, though labellar reflexion also contributes significantly. However, once again, there appears to be only partial separation of pollostantha from micrantha, demonstrating that azorica is the most morphologically distinct of the three species. The third and fourth axes, which also combined to yield a diagonal relationship, served to separate micrantha from pollostantha and azorica. The third axis is determined by characters that distinguish pollostantha from micrantha, including labellum width, position of lateral sepal and labellar reflexion.

Figure 19 Principal coordinates plots for the first two axes for two different combinations of taxa and characters (parenthetic figure indicate the percentage of the total variance accounted for by each axis).

(A) Three taxa, all characters. (B) Two taxa, all characters. For characters contributing to each axis see Tables 2B and 2D.

A further analysis of these individuals omitted all vegetative characters (Fig. 18B), on the grounds that they are on average more vulnerable to ontogenetic and ecophenotypic variation (e.g., Bateman & Rudall, 2011). The remaining 21 variables strengthened the first axis relative to the second (Table 2C). Predictably, the first axis separated P. azorica from the remaining species, once again primarily on the basis of labellum, petal and sepal lengths, supported other floral dimensions. The second axis not only separated P. micrantha from P. pollostantha but also distinguished the two suspected hybrid plants, though it did not place them as morphologically intermediate to their parents. This axis similarly resembled the second axis from the full matrix, being dictated by labellum width, position of lateral sepal and labellar reflexion. Lower-order axes represented single non-diagnostic characters and served no taxonomic purpose.

Two taxa

We then further simplified the analysis by removing the 20 plants of P. azorica measured plus the associated holotype, in order to better assess the much-debated relationship between P. pollostantha and P. micrantha. Although the first two axes were both weakened, they again yielded diagonal distributions of conspecific plants; however, they now fully separated the two remaining species (Fig. 19B). Also, the two hybrid plants were placed between the two putative parents, albeit substantially closer to micrantha than to pollostantha. The first axis was once again determined by labellum, petal and sepal lengths, supported by spur, leaf, bract and ovary dimensions (Table 2D). The second axis was dictated by labellum width and reflexion. Lower-order axes, based on leaf and bract length and spur curvature respectively, again lacked obvious significance.

Single taxon

Finally, analyses were conducted at the level of single species (results not shown), the focus of interpretation consequently shifting downward by one demographic level from species to single-island populations. Our intentions were to identify any subtle inter-island differences in morphology and also to place geographically two historical holotypes, which were not attributed to particular Azorean islands by the Hochstetters (father and son) or by Seubert.

Although the 142 plants of P. pollostantha sampled from eight islands yielded strong first and second axes, few island-related patterns were evident. The first axis did separate Corvo from Faial plants, but this is not surprising as both islands were represented by only single sampled localities. The second axis partially separated individuals from Faial and São Miguel from those found in Terceira and Santa Maria. The first axis gave unusual prominence to bract dimensions, admixed with dimensions of floral parts plus ontogenetically variable characters such as leaf and flower numbers and stem dimensions (Table 2E). The second axis continued the vigour theme by summarising lengths of bracts, leaves, stem and inflorescence. The holotype of P. pollostantha was unhelpfully placed in a location on the plot that was less than 0.01 of a multivariate unit distant from individuals sampled on five of the nine Azorean islands, thus eliminating any hope of inferring its island of origin (Schlechter, 1920, implied that the holotype originated from Pico, but we can find no historical evidence to support this assertion).

A further analysis using 54 plants of P. micrantha sampled from five islands also yielded two well-supported coordinates, but they more closely resembled each other in percentage of total variance explained and emphasised characters that suggest little structure to the data. The first axis was dominated by an antagonistic relationship between spur curvature and stem diameter, and the second axis by a combination of lateral sepal position and basal bract length (Table 2F). This axis served primarily to separate from the remainder the two plants measured on Flores. The plot suggested that the nineteenth century holotype of P. micrantha most closely resembled plants from São Miguel or Pico, though ad hoc correspondence is quite likely in the case of Pico, as the island contributed 20 of the 54 plants analysed.

Univariate analyses

Table 3 gives mean, sample standard deviation and coefficient of variation values for all 38 morphometric characters measured in each of the three Azorean species of Platanthera. Data are also given for five individual plants of particular interest: two putative hybrids between P. pollostantha and P. micrantha (from the Trilho Topo locality, near the eastern end of São Jorge) and the holotypes of each of the three species.

Table 3 Population means, sample standard deviations (SSD) and coefficients of variation (CV, %) for 38 morphometric characters.

Data were recorded from 141 plants of Platanthera pollostantha, 53 plants of P. micrantha and 20 plants of P. azorica, together with the holotype specimens of the three species and two putative hybrid plants from the Trilho Topo locality on São Jorge.

	Species	length
lab	width
lab	reflexion
lab	pigment
lab	extent
pigm
lab	length
spur	width
mouth
spur	width
halfw
spur	curvature
spur	length
ovary	
Mean	pollostantha	2.86	2.08	1.35	1.94	74	3.12	0.77	0.82	4.97	8.35	
SSD		0.7	0.34			25.2	0.41	0.25	0.15		1.84	
CV(%)		24.5	16.3			34.1	13.1	32.5	18.3		22	
												
Mean	micrantha	4.61	1.57	0.02	1.64	87.7	7.27	0.75	0.88	4.83	11.9	
SSD		0.84	0.27			20.1	0.97	0.14	0.16		1.75	
CV(%)		18.2	17.2			22.9	13.3	18.7	18.2		14.7	
												
Mean	azorica	8.32	2.41	3.1	1.8	76.5	9.51	1.36	1.12	4.7	14.0	
SSD		0.98	0.24			5.9	0.89	0.55	0.25		1.4	
CV(%)		11.7	10			7.7	9.4	40.4	22.3		10	
												
Holotypes	micrantha	2.3	1.5	1	NM	NM	2.4	0.5	0.6	5	8.5	
	azorica	4.2	1.1	0	NM	NM	6.5	0.7	0.8	5	11	
	adelosa	7.2	1.4	3	NM	NM	7.9	0.9	0.6	4	12	
												
Hybrid (1)	pollost. × micr.	2.68	1.32	0	2	80	4.8	0.4	0.5	4	6.08	
Hybrid (2)	pollost. × micr.	1.78	1.27	0	2	80	4.48	0.5	0.91	4	8.37	
	Species	pos
lat
sepal	length
lat
sepal	width
lat
sepal	length
lat
petal	colour
lat
petal	length
col	width
col	width
stigma	length
poll	distance
visc	
Mean	pollostantha	1.04	3.38	2.32	2.27	1	1.26	1.39	0.84	0.91	0.83	
SSD			0.7	0.35	0.59		0.27	0.25	0.29	0.2	0.17	
CV(%)			20.7	15.1	26		21.4	18	34.5	22	20.5	
												
Mean	micrantha	0.13	4.94	2.77	3.16	1	1.46	1.54	1.17	1.15	1.01	
SSD			0.68	0.44	0.47		0.26	0.25	0.21	0.17	0.2	
CV(%)			13.8	15.9	14.9		17.8	16.2	17.9	14.8	19.8	
												
Mean	azorica	1	8.19	3.64	5.7	1	3.16	3.49	2.67	2.49	3.1	
SSD			0.92	0.46	0.85		0.51	0.41	0.5	0.54	0.37	
CV(%)			11.2	12.6	14.9		16.1	11.7	18.7	21.7	11.9	
												
Holotypes	micrantha	1	2.7	1.6	1.9	NM	1.2	1.2	NM	0.8	NM	
	azorica	0	4.5	2.2	3.8	NM	1.2	1.4	NM	1	NM	
	adelosa	1	8	3.1	4.5	NM	2.7	3.2	NM	2	NM	
												
Hybrid (1)	pollost. × micr.	0	2.67	1.8	2.83	1	0.9	1	0.6	0.5	1	
Hybrid (2)	pollost. × micr.	0	3.5	2.14	2.6	1	1.2	0.7	0.6	0.7	0.8	
	Species	distance
apices	length
stamin	mean
cell
diam
bract	mean
cell
shape
bract	width
floral
bract	length
floral
bract	length
basal
bract	stature
stem	length
spike	no.
flowers	
Mean	pollostantha	0.47	0.47	53	1	3.47	10.28	27.07	24.8	77	40	
SSD		0.15	0.11	12		1	3.37	19.4	8.4	28	18	
CV(%)		31.9	23.4	22.6		28.8	32.8	71.7	33.9	36.4	45	
												
Mean	micrantha	0.64	0.53	47	1	4.94	13.44	21.17	31.7	109	60	
SSD		0.13	0.2	8		1.03	3.21	13.6	8.8	40	30	
CV(%)		20.3	37.7	17		20.9	23.9	64.2	27.8	36.7	50	
												
Mean	azorica	2.2	0.83	51	1	5.37	18.07	28.31	20.1	85	18	
SSD		0.3	0.43	8		0.9	3.52	8.35	4.9	23	5.5	
CV(%)		13.6	51.8	15.7		16.8	19.5	29.5	24.4	27.1	30.6	
												
Holotypes	micrantha	NM	NM	NM	NM	3.2	13	16	30	65	28	
	azorica	NM	NM	NM	NM	4.2	15	19	25	93	68	
	adelosa	NM	NM	NM	NM	4.2	20	28	21	65	10	
												
Hybrid (1)	pollost. × micr.	0.8	0.5	48	1	2.19	7.03	30.02	26	65	25	
Hybrid (2)	pollost. × micr.	0.5	0.6	NM	NM	1.95	5.45	30	27	95	41	
	Species	stem
diam	no.
non
sheathing
leaves	no.
sheathing
leaves	width
longest
leaf	length
longest
leaf	length
max
width/
length	petiole
developm	angle
ground			
Mean	pollostantha	2.93	4.12	2.13	31.4	105.1	61.58	0.12	2.17			
SSD		0.99	1.49	0.52	15.1	36.6	9.01					
CV(%)		33.8	36.2	24.4	48.1	34.8	14.6					
												
Mean	micrantha	3.69	3.25	2.32	54.4	125	58.8	0.04	2.04			
SSD		1.21	1.28	0.61	15.4	36.5	4.9					
CV(%)		32.8	39.4	26.3	28.3	29.2	8.3					
												
Mean	azorica	3.82	1.75	1.85	40.1	111	59.8	0.05	2			
SSD		1.01	0.72	0.37	11.8	32.1	5.2					
CV(%)		26.4	41.1	20	29.4	28.9	8.7					
												
Holotypes	micrantha	2.7	5	2	46	105	55	0	2			
	azorica	3.2	5	2	46	110	50	0	2			
	adelosa	2.5	0	2	26	93	50	0	2			
												
Hybrid (1)	pollost. × micr.	1.98	2	2	30	100	NM	0	2			
Hybrid (2)	pollost. × micr.	3	2	2	36	110	NM	0	2			
Notes.

NM Not measurable

Potentially diagnostic scalar characters were summarised as histograms: selected for presentation here are histograms for labellum reflexion, lateral sepal position, number of sheathing leaves and number of non-sheathing leaves (Fig. 20). Metric and meristic characters of particular interest were plotted together in pairwise combinations to yield scatter-diagrams of individual plants. Examples shown here are length versus width of gynostemium and viscidial separation versus pollinarium length (Figs. 21A and 21B), labellum length versus labellum width and labellum length versus spur length (Figs. 22A and 22B), ovary length versus spur length and leaf length versus leaf width (Figs. 23A and 23B), and lastly, labellum length versus lateral sepal length and inflorescence length versus number of flowers in inflorescence (Figs. 24A and 24B). Where appropriate, linear regressions were plotted for each of the three Azorean species of Platanthera (Figs. 23 and 24A).

Figure 20 Univariate histograms of plants of the three Azorean Platanthera species.

(A) Labellum position viewed laterally (0 = strongly decurved, 1 = slightly decurved, 2 = vertical, 3 = slightly recurved, 4 = strongly recurved). (B) Lateral sepal position as viewed vertically (0 = near-vertical, 1 = substantially below horizontal, 2 = more-or-less horizontal). (C) Number of sheathing leaves. (D) Number of non-sheathing (bracteoidal) leaves. Letters indicate the conditions for these characters inferred in the holotypes of the three species.

Figure 21 Bivariate scatter-diagrams of plants of the three Azorean Platanthera species.

(A) Gynostemium length versus gynostemium width. (B) Distance separating paired viscidia versus pollinarium length (note that the three holotype specimens could not be measured with adequate accuracy for these characters).

Figure 22 Bivariate scatter-diagrams of plants of the three Azorean Platanthera species.

(A) Labellum length versus labellum width. (B) Labellum length versus spur length.

Figure 23 Bivariate scatter-diagrams of plants of the three Azorean Platanthera species.

(A) Ovary length versus spur length. (B) Leaf length versus leaf width. Both graphs include linear regressions for each species; (A) also shows three arbitrary threshold ratios for spur length over ovary length (dashed lines).

Figure 24 Bivariate scatter-diagrams of plants of the three Azorean Platanthera species.

(A) Labellum length versus lateral sepal length, including linear regressions for each species. (B) Inflorescence length versus number of flowers.

Each of these 16 figured plots provides substantial discrimination between at least two of the three Azorean Platanthera species. The significance of the discrimination that is revealed, and of the characters that underlie that discrimination, are considered in the following detailed Discussion.

Discussion

Phylogenetic and evolutionary context

Genus-level assignment

Circumscription of genera within Orchidaceae tribe Orchideae has been much debated (reviewed by Vermeulen, 1947; Bateman, Pridgeon & Chase, 1997; Bateman et al., 2003; Bateman et al., 2009). The boundary separating Habenaria from Platanthera has fluctuated greatly between taxonomic treatments through the centuries. However, most authors at least agreed that the two genera were closely related, due primarily to their broadly similar floral morphologies.

Each nineteenth century account of the Azorean species assigned them to Habenaria (Seubert & Hochstetter, 1843; Seubert, 1844; Drouet, 1866; Watson, 1870; Trelease, 1897), before Schlechter (1920) correctly recognised that features of the gynostemium demonstrated that the Azorean species belonged to Platanthera. These characteristics were made more explicit by Schlechter (1992), who noted the greater fusion of organs in the gynostemium and the comparatively subdued rostellum (also, Old World representatives of the two genera can be distinguished by the deeply trilobed labellum of most Habenaria species). Nonetheless, a minority of authors continued to assign the Azorean species to Habenaria (e.g., Palhinha, 1966; Sjögren, 1973). Finally, DNA data demonstrated that the molecular divergence of Platanthera from Habenaria is considerably greater than the corresponding morphological divergence (e.g., Bateman et al., 2003), placing the two genera in different subtribes and thereby unambiguously deciding the long-debated issue of their relationship. This insight then provided the necessary context for Bateman et al. (2009) to sequence samples of the Azorean ‘Habenaria’ species and demonstrate unequivocally that they belong to the genus Platanthera.

Origin(s) of the Azorean Platanthera lineage

Previous commentators have uniformly agreed that, if two species of Platanthera did indeed occur on the Azorean archipelago, they were closely related; there was also an underlying assumption (more often implicit than explicit) that both species represented a single immigration event of Platanthera seed from a particular continental source. The majority of commentators believed that this source lay to the northwest rather than the east. For example, Delforge (2003, 106–7) argued that “morphological analysis suggests that their closest relative is probably Platanthera hyperborea, a North American subarctic species reaching Greenland and Iceland” and hence concluded that, when taxonomically listing European orchids, “P. micrantha and P. azorica should be placed directly after P. hyperborea and before P. obtusata, rather than before P. bifolia” (both quotes translated by us from the original French text).

Earlier, Schlechter (1920, 377) had reached a less confident conclusion, stating that “investigation of the flowers of both species has shown that we have before us typical Platanthera species, but which are not sufficiently closely related to the European or the North American species that they could be derived from them. It is instead a question of completely isolated types that are well understood as relics, many examples of which we also find in Madeira and the Canary Islands” (translated from German). In other words, Schlechter viewed the Azorean Platanthera lineage as having occupied the islands for so long that it was no longer feasible to identify its phylogenetic relationships.

The questions of both the phylogenetic position of the Azorean species and their degree of divergence from their closest relatives were unequivocally answered by the nuclear ribosomal Internal Transcribed Spacer (ITS) phylogeny of Bateman et al. (2009; see also Bateman, James & Rudall, 2012). Their tree demonstrated large molecular disparities that readily distinguish between the Azorean species P. pollostantha plus P. micrantha and several other species-groups of Platanthera from Eurasia and North America that bear small, green flowers (Bateman et al., in press). The molecular distance is particularly great relative to the dominantly North American P. hyperborea complex of diploid and polyploid species (e.g., Sheviak, 2002), which extends geographically as close to the Azores as Newfoundland and Iceland. Moreover, the P. hyperborea complex was the origin of the exceptionally rare species P. holochila, which speciated in the scrubby laurisilva-like cloud forests of the even more remote Hawaiian islands (Lauri, 2010; Bateman et al., in press).

Instead, the Azorean species showed close genetic similarity to the widespread Eurasiatic P. bifolia complex. Remarkably, a single ribotype is dominant in all of the species of Platanthera recorded in and around the Mediterranean (i.e., P. bifolia, P. chlorantha, P. algeriensis, P. holmboei). The two predominant ribotypes found among the Azorean taxa show them to be derived relative to their mainland cousins, and P. micrantha to be derived relative to P. pollostantha and P. azorica (Bateman et al., in press). In retrospect, placement of the Azorean species within the P. bifolia-chlorantha aggregate rather than the P. hyperborea aggregate could have been predicted from the morphology of its tubers alone, which are fusiform (Fig. 4E), contrasting with the filiform tubers that characterise the majority of Platanthera species, including those of the hyperborea group (cf. Sundermann, 1980: Fig. 211; Efimov, 2011: Fig. 4). Taken together, these data deliver a coup de grace to the competing hypotheses of both Schlechter (1920) and Delforge (2003); the Azorean Platanthera lineage(s) actually reached the islands comparatively recently rather than being deeply relictual, and unquestionably emigrated from the Old World rather than from the New.

We envisage a single migration from within the P. bifolia-chlorantha aggregate – most likely by westward transport of dust-seeds from a population in the Mediterranean (perhaps of P. algeriensis in North Africa or Iberia, though we currently lack any strong evidence to support such an inference: Bateman et al., in press). Such long-distance airborne dispersal is, by definition, likely to entail both an intense genetic bottleneck and a strong founder effect through the immigrant being in at least some ways genotypically and phenotypically unrepresentative of the source population (e.g., Bateman & Devey, 2006). And once it has successfully established its first colony on the island, the small founder population, essentially free of a serious risk of further immigration of conspecific seeds, will be especially vulnerable to genetic drift (e.g., Tremblay et al., 2005). This combination of genetic effects creates an ideal environment for the anagenetic speciation that apparently explains the origins of most of the Macaronesian orchids (Bateman et al., in press) and is likely to account for the origin of either one or more likely two of the three Azorean Platanthera species (see ‘Species-Level Relationships’). This initial anagenetic shift was most likely followed by at least one cladogenetic speciation event on the islands, the overall phylogenetic picture being further confused by extensive inter-island migrations (Fig. 1) involving at least two of the three Azorean species (cf. Bateman & Devey, 2006; Roberts & Bateman, 2006; Bateman, 2012; Bateman et al., in press).

Irrespective of which of the two hypotheses of species relationships outlined below is the more accurate, it seems likely that the three Platanthera species still lie within, or at best only recently escaped from, the period immediately following speciation, when levels of phenotypic divergence inevitably greatly exceed levels of genotypic divergence. This ubiquitous evolutionary stage was termed the ‘genetic divergence lag’ by Bateman (e.g., Bateman, 2011; Bateman, James & Rudall, 2012).

More generally, our molecular data support the major conclusions of the recent synthesis of Azorean plant origins published by Schäfer et al. (2011). They argued that earlier assertions of under-representation of endemic species in the Azores (e.g., Carine & Schäfer, 2010) were premature, cryptic species being more numerous than was originally thought, and that the flora of the archipelago remains under-researched by evolutionary biologists.

Species-level relationships and morphological disparities

The precise relationships among the continental European and Azorean Platanthera species remain contentious, not least because the implications of data from plastid microsatellites and from morphology appear contradictory. Specifically:

The molecular data clearly identify the P. bifolia-chlorantha aggregate as having given rise to the Azorean lineage – or, stated more accurately, they identify the Azorean lineage as being an integral part of the P. bifolia-chlorantha aggregate, despite their many morphological contrasts. The detailed, population-level analyses of plastid haplotypes by Bateman et al. (in press), comparing Azorean with mainland species (notably the central Mediterranean species analysed by Pavarese et al., 2011), indicate approximately equal probabilities of single or multiple origins of the Azorean Platanthera lineage. In terms of species of origin, all of the Mediterranean species of Platanthera yielded individuals placed within three parsimony steps of at least one Azorean plant. Similar ambiguities plague attempts to use the haplotypic data to infer the identity of the first-formed Azorean species; P. pollostantha and P. micrantha appear equally likely candidates from plastid evidence. Given that the ITS ribotypes indicate that P. micrantha is derived relative to P. pollostantha and P. azorica (Bateman et al., 2009; Bateman, James & Rudall, 2012; Bateman et al., in press), when considered together, haplotypes and ribotypes suggest that P. pollostantha was the first-formed Azorean species. Given that all of the potential mainland ancestors are large-flowered, any of the resulting evolutionary scenarios requires radical reduction in flower size to generate the small-flowered P. pollostantha and P. micrantha. They also require a reversed radical expansion in flower size, as well as restoration of papillae within the labellar spur in order to generate the seemingly atavistic large-flowered morphology of P. azorica (Bateman et al., in press).

Admittedly, our morphological observations imply a substantially different story. The multivariate analysis of morphometric data for all seven species (Fig. 18A) shows that the three Azorean species have approximately equal morphological similarities to P. bifolia s.s. (a species distributed only sporadically through Iberia) and P. algeriensis (a species that occurs in both Iberia and northwest Africa, but one that, despite our best efforts, is under-sampled in this morphometric data-set). However, P. azorica is revealed to be morphologically similar (though not identical) to the eastern Mediterranean P. holmboei. When multivariate comparison is reduced to the three Azorean species (Fig. 19A), a substantial morphological discontinuity is also seen to separate P. azorica from P. micrantha and P. pollostantha, which appear to overlap morphologically. However, this appearance of close similarity between P. micrantha and P. pollostantha is somewhat deceptive, reflecting the fact that their morphological divergence involved a somewhat different suite of characters from those that distinguish the other five European species of Platanthera (Table 2). Once the ecophenotypically malleable vegetative characters are removed from the analysis (Fig. 18B), or comparison is reduced to just P. pollostantha versus P. micrantha (Fig. 19B), individuals of the two small-flowered species are readily distinguished morphologically.

However, ample evidence has accumulated to suggest that both overall flower size and strength of green pigmentation are highly evolutionarily labile, and so phylogenetically homoplastic (Bateman et al., 2009), within the Platanthera s.l. clade. We therefore turned our attention to the details of the gynostemium (Figs. 11–17: see also Figs. 4 and 15 of Bateman, James & Rudall, 2012). Here, it is very clear that the gynostemium morphology of P. azorica is remarkably similar to that characterising P. chlorantha and its segregates P. holmboei and P. algeriensis s.l., whereas the gynostemia of P. micrantha and P. pollostantha more closely resemble that of P. bifolia. These observations suggest that not one but two mainland emigrés colonised the Azores; if so, P. azorica would have resulted from anagenetic miniaturisation of an immigrant ancestor resembling P. chlorantha (or, more likely, P. algeriensis or P. holmboei), whereas either P. pollostantha or P. micrantha would have originated from anagenetic miniaturisation of an immigrant ancestor of P. bifolia – an event that was followed by the cladogenetic divergence of P. micrantha from P. pollostantha (or vice versa). An alternative hypothesis of relationships, more consistent with the molecular data, requires two remarkable convergences of gynostemium morphology – first from the chlorantha-type to bifolia-type morphology to generate the initial pollostantha-micrantha lineage, and then back to the chlorantha-type morphology to produce P. azorica (Bateman et al., in press).

Inferred speciation mechanisms

As detailed by Bateman et al. (in press), mycorrhizal specificity appears to have played an important role in the origins of the Azorean species, though a contribution to speciation from divergence of pollinator spectra also seems likely. Here, we focus on the contribution of phenotype to speciation in the group.

Bateman, James & Rudall (2012) emphasised the evolutionary significance within the P. bifolia-chlorantha aggregate of developmental shifts – both allometric and non-allometric – in determining the relative and absolute dimensions of floral organs. In particular, they noted in the sizes of many structures a ratio approximating 2:3 between P. bifolia s.s. and P. chlorantha. Intriguingly (but probably coincidentally), there is also an average 2:3 ratio in the majority of floral structures between P. pollostantha and P. micrantha, and between P. micrantha and P. azorica.

Both the molecular and morphological hypotheses of relationship outlined above require one initial speciation event through radical reduction in flower size, from either a bifolia-like or chlorantha/algeriensis/holmboei-like ancestor, respectively, to a phenotype most likely resembling P. pollostantha, followed by a far less radical transition to generate P. micrantha. The molecular hypothesis of relationship then requires an equally radical expansion in size to produce P. azorica, whereas the morphological hypothesis suggests a second immigration event, this time of a member of the P. chlorantha group (perhaps resembling P. holmboei) followed by a (more modest) decrease in flower size to generate P. azorica.

Irrespective of any preferred scenario of species-level relationships, none of these postulated phenotypic transitions associated with speciation is strictly allometric in all characters (Fig. 25). For example, P. azorica is approximately one third smaller than P. algeriensis in the majority of characters depicted in Fig. 25B but equals that species in labellum width and lateral petal length, whereas average spur length is halved. And compared with P. bifolia, P. micrantha has disproportionately large reductions in spur and labellum lengths but has retained the putatively ancestral widths of the lateral sepal and gynostemium (Fig. 25A). Platanthera pollostantha has apparently experienced even greater spur and labellum length reductions, but has retained a relatively wide labellum comparable with that of P. bifolia.

Figure 25 Spider-diagrams of eight metric variables (mm) that yield mean values capable of distinguishing between the three Azorean species of Platanthera and their presumed ancestral species.

(A) Platanthera bifolia and its putative Azorean descendants, P. micrantha and P. pollostantha. (B) Platanthera chlorantha, P. algeriensis, P. homboei and their putative Azorean descendant, P. azorica (but see Bateman et al., in press).

As a result of these presumed paedomorphic transitions, the gynostemium appears to have been reduced as radically as is feasible without becoming severely dysfunctional. The entire structure has decreased greatly in width and especially in length, requiring straightening of the anther locule and straightening, shortening and flattening of the caudicles. The number of rows (and total number) of massulae has greatly diminished, though in contrast, there has been little reduction in the size of individual massulae. Most strikingly, the stigmatic surface has decreased greatly in extent, reversing the radical expansion of the stigma inferred by Bateman, James & Rudall (2012) to have driven the earlier transition from P. bifolia to P. chlorantha – one that presumably occurred in mainland Eurasia.

With the exception of a possible ancestor–descendant relationship between P. pollostantha and P. micrantha, these hypothesised phenotypic shifts appear radical and unsubtle, presumably reflecting relatively simple developmental-genetic underpinnings (cf. Bateman, James & Rudall, 2012). The degree to which these changes confer adaptive advantage on the resulting lineages remains entirely speculative (cf. Tremblay et al., 2005; Bateman, 2012). Further progress in understanding the mechanisms of speciation responsible for the three Azorean Platantheras must await a clearer molecular phylogenetic framework, together with studies within each species of several additional factors; these should include (1) chromosome number, structure and genome size (predicted on current evidence to be very similar), (2) pollinator attractants (especially the biochemistry of their contrasting fragrances), (3) the identities of their insect pollinators, (4) the frequencies of seed-set within individual plants, (5) the frequencies of autogamy and/or geitonogamy within populations, and, most importantly, (6) the genetics underpinning development of the floral organs, especially key elements of the gynostemium (cf. Bateman, James & Rudall, 2012; Rudall, Perl & Bateman, 2013).

Species circumscription and recognition

Diagnostic characters

As noted in previous reviews (Rückbrodt & Rückbrodt, 1994; Delforge, 2003), the majority of authors who have expressed taxonomic opinions since P. pollostantha and P. micrantha were first described (as P. micrantha and P. azorica, respectively) by Seubert (1844) have accepted both taxa as full species (Drouet, 1866; Watson, 1870; Trelease, 1897; Schlechter, 1920; Schlechter, 1923; Keller & Schlechter, 1928; Palhinha, 1966; Sjögren, 1973; Baumann & Künkele, 1988; Buttler, 1991; Rückbrodt & Rückbrodt, 1994; Schäfer, 2002; Delforge, 2003; Kreutz, 2004; Delforge, 2006; Tyteca & Gathoye, 2012; reputedly also Frey & Pickering, 1975; Frey, 1977). We find it particularly instructive that Sundermann (1980) reversed his earlier decision (Sundermann, 1975) and chose to recognise two distinct species of Azorean Platanthera, as his monograph in general constitutes the most extreme example of taxonomic ‘lumping’ at species level ever attempted for the European orchid flora. In contrast, the bizarre decision of Kränzlin (1897–1904) to synonymise both Azorean Platanthera species with unrelated North American species was thoroughly refuted by Schlechter (1920).

Nonetheless, other authors chose to treat the second putative species as being synonymous with the first (Soó, 1930–1940; Hansen, 1972; Rasbach & Rasbach, 1974; Sundermann, 1975; Sjögren, 1984; Hansen & Sunding, 1993; Sjögren, 2001). A few other authors listed P. micrantha (as “P. azorica”) but expressed doubt regarding its biological reality (e.g., Williams, Williams & Arnott, 1979; Davies, Davies & Huxley, 1983). Yet others were more explicit, arguing that P. micrantha should be treated as a subspecies (Soó, 1930–1940) or variety (Webb, 1980) of P. pollostantha. Writing in the influential Flora Europaea, Webb (1980, 331) stated that “P. micrantha” was “variable, especially in length of spur; in most plants this is 2–3.5 mm, but in some it is 5–8 mm. The long-spurred variants have been distinguished as P. azorica Schlechter, loc. cit. (1920) (listed as Habenaria longibracteata [sic] Hochst.), but as variation in other characters shows little correlation [sic], and as there is no clear geographical of ecological separation, they are best treated as a variety of [P. micrantha]”. However, we have not been able to trace formal new combinations for ‘azorica’ at either subspecific or varietal rank; presumably, these authors were not sufficiently confident in the accuracy of their species circumscriptions. These contrasting taxonomic opinions provided part of the motivation for our morphometrically based systematic revision.

In fact, most of the 38 morphometric characters measured by us contributed to some degree towards distinguishing among the three Azorean endemic species. Exceptions to this rule were the size and shape of the cells marginal to the bracts. Bract-cell size proved to be useful for distinguishing between diploid and tetraploid species of the related orchid genus Dactylorhiza (e.g., Bateman & Denholm, 1983), but the near-uniformity observed here at the cellular level suggests that the three Azorean Platanthera species are probably reliably diploid, as are most Platantheras outside subgenus Limnorchis. As is usual in morphometric matrices describing European orchids, coefficients of variation are in most characters considerably higher for vegetative than for floral features (Table 3), reflecting the greater modifying influences of both ontogeny and ecophenotypy on vegetative organs (e.g., Bateman & Denholm, 1989; Bateman & Rudall, 2011). Unsurprisingly, it is the flowers that reliably provided the best diagnostic characters among the Azorean species. Nonetheless, coefficients of variation for floral characters were on average greater than in the many other groups of terrestrial orchids previously studied by us; we suspect that measuring errors contributed to these comparatively high values, reflecting both subtle differences between operators and decreased resolution caused by the unusually small sizes of some floral structures, notably the minute gynostemium (Figs. 11, 13 and 14).

Platanthera azorica is shown to be the most distinct of the three Azorean species in both the multivariate and univariate analyses (Figs. 18–24). It differs strongly or moderately from the two remaining species in 14 of the 21 variable floral characters measured (Table 3) – lengths of the labellum, spur, lateral petal and lateral sepal (also, to a lesser degree, its width). Even more striking are the differences in gynostemium dimensions – overall length and width, stigma width, pollinarium length, and distances separating both the viscidia and the pollinium apices. The gynostemium of P. azorica shows much closer similarity to those of continental P. chlorantha, P. algeriensis and especially P. holmboei than to those of the two remaining Azorean Platanthera species (Figs. 11–17). Microscopic examination also revealed a close similarity of P. azorica to P. holmboei/algeriensis/chlorantha in the distinctive morphology of the caudicle and viscidium (Figs. 11 and 15–17) (cf. Gölz & Reinhard, 1990; Bateman, James & Rudall, 2012). Returning to the morphometric characters, P. azorica also incurs the smallest coefficients of variation for most characters, presumably reflecting the fact that all of the plants measured were drawn not only from a single island but also from what we presume to be a single metapopulation. The most notable exception is the large coefficient of variation for diameter measured at the mouth of the spur, where its funnel shape most likely increased measuring error.

It is more difficult to distinguish morphologically between P. pollostantha and P. micrantha. The most effective distinguishing characters among those recorded morphometrically are labellum length (especially when combined with labellum width: Fig. 22A) and labellum reflexion, spur length and lateral sepal position (though comparison is made more difficult by the fact that the spur of P. micrantha often twists laterally, typically to the left as viewed from the entrance, and the lateral sepals of a few individuals are swept strongly backwards). Less reliable distinguishing characters include the lengths of lateral petals, lateral sepals (Fig. 24A) and ovaries (Fig. 23A), together with leaf width (Fig. 23B). The gynostemia of the two species are quite similar, though that of P. micrantha appears to be broader relative to its height (Figs. 11, 13 and 14). Moreover, the viscidia of P. micrantha are larger, more elongate and more clearly bipartite than those of P. pollostantha.

One taxonomically useful character that was not formally scored during our survey was flower colour (in the broadest sense of the term). All Eurasian Platanthera species lack substantial quantities of colourful anthocyanin pigments; their flower colours consequently simply vary from dark green (caused by the presence of relatively large numbers of chloroplasts) through to what is usually described as cream or white, but is in truth a translucent, essentially colourless condition that we believe merely reflects a comparatively low density of chloroplasts. A greater degree of translucency, especially evident in the lateral sepals when viewed in sunlight, helps to distinguish P. micrantha from the remaining Azorean species (Fig. 5C). The resulting colour, grading from pale green to cream, appears to reflect relatively thin lateral sepal laminae that are deficient in layers of mesophyll cells most likely to contain functional chloroplasts. Although the flowers of the two remaining Azorean species appear superficially to be a uniform green colour, detailed colour matching revealed a tendency for the labellum colour to shift very slightly from the blue towards the yellow end of the green spectrum in all three species. We suspect that the chloroplasts responsible for the green colour of the flowers are of more than ornamental function – the presence of apparently well-formed stomata in all perianth segments other than the labellum indicates that considerable photosynthetic and respirational activity occurs within the flower.

Several of those earlier authors who chose to recognise both P. pollostantha and P. micrantha at species level proposed small numbers of putatively diagnostic characters. The only previous author to offer detailed formal descriptions of both P. pollostantha and P. micrantha was Schlechter (1920). The five characters proffered as being diagnostic by Schlechter (1920), Sundermann (1980) and Rückbrodt & Rückbrodt (1994) included three that are shown by our data to be the most effective, specifically the contrasts in the position of the lateral sepals relative to the vertical, the orientation and direction of curvature of the labellum, and the length of the spur relative to that of the ovary. Sundermann (1980) and Delforge (2003) correctly added to these characters the contrast between the uniformly green flowers of P. pollostantha and the “whitish” sepals found in most individuals of P. micrantha. Other characters previously suggested as diagnostic have proven to be of little value, such as the ratios of ovary length to both spur length and bract length (cf. Schlechter, 1920; Rückbrodt & Rückbrodt, 1994). Moreover, distinctions between species in quantitative characters have often been exaggerated; for example, Buttler (1991) over-estimated the disparity in spur length between P. pollostantha and P. micrantha.

Ontogeny and ecophenotypy

No characters are more subject to contrasts in plant size/maturity (i.e., ontogeny) and the influence of the local environment (i.e., ecophenotypy) than flower number and especially inflorescence length, a character that in terrestrial orchids inevitably increases in magnitude as anthesis progresses. Together, these two parameters determine the density of the inflorescence (measured as flowers per centimetre: Bateman & Rudall, 2006), which is also influenced by flower size; once exceeding a certain density threshold that is dictated by flower shape and size, individual flowers would presumably suffer a serious decline in their ability to attract pollinators.

Thus, although the plot of inflorescence length versus flower number (Fig. 24B) may initially appear chaotic, it is nonetheless clear that P. azorica cannot condense its much larger flowers beyond ca 7 fls/cm – the lower threshold of density for the other two species, whose small flowers permit exceptionally compact inflorescences (means are 4.7 ± 1.4 for P. azorica versus 18.2 ± 7.8 for P. micrantha and 19.3 ± 7.9 for P. pollostantha: Table 3). The inflorescences of P. micrantha can often appear looser than those of P. pollostantha (Tyteca & Gathoye, 2012), but this impression is largely a consequence of their longer average ovary length (Table 3). In addition, it is clear that P. micrantha has the (presumably largely genetically determined) tendency to generate a larger maximum size of inflorescence than the other two species; at least one individual of P. micrantha from four of the five islands sampled possessed an inflorescence that exceeded 100 flowers and/or a length of 16 cm (Fig. 24B). Admittedly, P. pollostantha also rarely achieves similar dimensions (Tyteca & Gathoye, 2012). Nonetheless, it is evident that even the most labile morphometric characters can carry some taxonomically useful information.

The Azorean lineage(s) of Platanthera certainly originated from within the P. bifolia-chlorantha aggregate (Bateman, James & Rudall, 2012; Bateman et al., in press). Flowering plants of these mainland species have a characteristic configuration of leaves that consists of a pair of broadly ovate sheathing leaves located at the base of the stem, plus 1–6 (typically 3 or 4) bracteoidal leaves that are fairly evenly distributed along the section of stem that separates the basal leaves from the inflorescence. There is a radical difference in size between the basal and bracteoidal leaves, and only rarely does a single leaf intermediate in morphology emerge between the paired basal leaves and the tiny bracteoidal leaves. A broadly similar leaf configuration characterises the Azorean Platanthera species. Most plants of all three species possess two sheathing leaves, though plants of P. micrantha occasionally possess three such leaves and ca 15% of plants of P. azorica flower successfully when possessing only one such leaf (Fig. 20C) – a similar percentage to that reported for P. chlorantha by Bateman, James & Rudall (2012). Interestingly, the number of non-sheathing, bracteoidal leaves (Fig. 20D) helps to distinguish P. azorica (2 or, less frequently, just 1 such leaf) from the other two species; P. micrantha has 1–6 such leaves (typically 2–4) and P. pollostantha has 2–8 (typically 3–5). In addition, the distinction between the two categories of leaf tends to be less clear-cut in the Azorean taxa (or at least in P. pollostantha and P. micrantha) than in their potential antecedents in mainland Europe; there appears to be a cline in the development of leaf-like organs that runs from the base of the stem to the apex of the inflorescence (Figs. 4–6).

These observations illustrate two broader principles. Firstly, once again, a character strongly influenced by ontogeny nonetheless offers some value for diagnosing species. Secondly, that diagnostic value would have been less evident had we failed to impose the initial distinction between basal and bracteoidal leaves and had instead lumped them into a single category, specifically total leaf number.

Regression slopes are more informative than arithmetic ratios

Sundermann (1980, 207) argued that P. pollostantha has “bracts usually much shorter than the ovary”, whereas P. micrantha has “bracts longer than the ovaries”. In fact, approximately one third of individuals of each of these species have floral bracts shorter than the ovaries whereas the remaining two thirds are longer. In contrast, bracts basal to the inflorescence are reliably substantially longer than the ovaries in both species, grading into the bracteoidal leaves beneath. Presumably, Sundermann simply lacked the data to test his hypothesis that this character is diagnostic.

The downside of using as taxonomic characters raw ratios is better illustrated by the bivariate scatter-diagram that plots ovary length against spur length (Fig. 23A). Seubert (1844) claimed that the spurs of “P. micrantha” (our P. pollostantha) were “half the length of the ovary”. However, as is usually the case between two metric characters, the regression lines for the three Platanthera species do not pass through the origin of the graph, and so their relationship cannot be satisfactorily represented as a simple arithmetic ratio. Moreover, in the case of P. pollostantha, longer ovaries do not correlate with longer spurs, perhaps because in this species the spurs are so much shorter than the ovaries that there is little if any functional linkage between the two structures. In other words, spur length has seemingly fallen below a critical threshold of developmental correlation. Thus, in plants of P. pollostantha with ovaries 14.5 mm long, the spur is predicted by our data to be one quarter the length of the ovary, whereas in conspecific plants with ovaries a mere 4.5 mm long the spur is predicted to be two-thirds of the length of the ovary (Fig. 23A). Admittedly, ratios are more consistent in P. micrantha and P. azorica – in both cases, spurs vary between one half and three-quarters the length of the ovary – but again the relationship depends to a considerable degree on absolute values for each parameter, not just relative values.

The plot of ovary length versus spur length (Fig. 23A) makes an interesting contrast with that for labellum length versus lateral sepal length (Fig. 24A), where the necessity of packaging all of the perianth segments into a cohesive bud has encouraged a strong positive correlation between these two metric dimensions. That correlation is preserved by, and of a similar strength within, all three Azorean Platanthera species. For each species the regression line has a similar slope, attracts a similar R2 value, and passes fairly close to the origin of the graph.

Regression and the recognition of non-flowering plants

In any one year and within any one population, the vast majority of Azorean butterfly-orchids fail to flower; indeed, populations are occasionally encountered in which no individual has flowered. Thus, it would be immensely helpful if vegetative characters could be found that alone were sufficiently diagnostic to at least distinguish between the two more widespread – and often co-existing – endemics, P. pollostantha and P. micrantha. Unfortunately, no vegetative micromorphological characters were found that showed taxon-specific differences, and leaf shape was reliably orbicular to obovate; leaves of all three species were on average widest at a point approximately 60% along their lengths from the base. Moreover, leaves that possessed well-developed peduncles and/or were held closer to the vertical than the horizontal were clearly ecophenotypically moulded, being largely confined to a few plants of P. pollostantha that occurred in dense, dwarf heathland dominated by Calluna.

The most effective vegetative criterion for distinguishing between P. pollostantha and P. micrantha proved to be the relationship between leaf length and leaf width (Fig. 23B). Using the intermediate P. azorica regression line as a convenient threshold between P. pollostantha (below the line) and P. micrantha (above the line) allows successful identification of 83% of the plants measured. Indeed, the success rate would be considerably higher were it not for the existence of a few plants of P. pollostantha bearing unusually wide leaves (hence the relatively low R2 value of 0.18: Fig. 23B).

Morphological and molecular recognition of hybrids

Two suspected hybrids between P. pollostantha and P. micrantha, possibly the product of a single cross-pollination event, were found at the Trilho Topo locality on São Jorge by one of us (MM). Although their identification as hybrids is not rendered certain by either the morphometric or the molecular data, it is strongly supported by both datasets (cf. Bateman et al., in press). The multivariate analyses (e.g., Fig. 19B) clearly place the two plants closer to P. pollostantha than to P. micrantha – indeed, the hybrids resemble P. pollostantha in the majority of characters that help to discriminate between the two species (Table 3). However, they more closely resemble P. micrantha in their narrow and deflexed labella, near-vertical presentation of the lateral sepals, and small number of non-sheathing leaves (in this character they actually most closely resemble P. azorica, although its sole known locality is ca 20 km distant). In addition, the hybrids are intermediate between the parents in spur length and petal length.

Having performed detailed morphometric analyses on several genera of Eurasian orchids, Bateman (e.g., Bateman & Hollingsworth, 2004; Bateman, Smith & Fay, 2008) argued that, as a general principle, hybrids resemble morphologically their ovule-parent more closely than their pollen-parent. Thus, we predicted that P. pollostantha would prove to be the ovule-parent of these two hybrid plants. However, our analysis of plastid haplotypes (Bateman et al., in press) – by definition inherited solely from the ovule parent – showed that the two hybrids share the haplotype most typical of P. micrantha, which was found in all three individuals of P. micrantha analysed from Trilho Topo. In contrast, all five of the individuals of P. pollostantha analysed from this locality presented the haplotype most typical of that species. This result strongly suggests that P. micrantha was the ovule parent of the two hybrid plants, despite their closer morphological resemblance to P. pollostantha (contra Bateman & Hollingsworth, 2004).

Achieving reproductive isolation in sympatry

Autogamy and Baker’s Law

As a general rule, terrestrial orchid species that offer a substantial nectar reward entertain wider spectra of pollinators than do deceitful species (e.g., Neiland & Wilcock, 1998; Cozzolino & Widmer, 2005; Claessens & Kleynen, 2011). In the case of mainland species of the P. bifolia-chlorantha aggregate, this rule is certainly followed; Claessens & Kleynen (2011) listed 29 species of 16 genera of Lepidoptera as pollinating P. chlorantha and 19 species of 12 genera of Lepidoptera as pollinating P. bifolia; moreover, eight of those butterfly and moth species are reputedly shared by the two species. In contrast, we are not aware of any reports of pollination observations relating to any of the Azorean Platanthera species – though neither are we aware of any concerted attempts to obtain such observations.

In his account of these orchids, Delforge (2003) argued that the pollen masses of both P. pollostantha and P. micrantha are sufficiently friable for the pollen grains to fall from the pollinia onto the stigmatic surface below, and that these plants are therefore most likely at least facultatively autogamous (a suggestion repeated – at least for P. pollostantha – by Tyteca & Gathoye, 2012). This is not an unreasonable prediction, as a relatively high percentage of ocean island endemic species become autogamous, presumably to free them from reliance on what will at best be a limited spectrum of potential pollinator lineages reaching these wind-blown islands from mainland sources (e.g., Roberts & Bateman, 2006). Nonetheless, we suspect that pollination of all three Platanthera species is most commonly enacted by comparatively small moth species, even though we failed in our brief attempts to net pollinaria-bearing moths from their likely refugia on the underside of leaves of shrubs surrounding flowering Platantheras. Five lines of evidence – admittedly all circumstantial – led us to the conclusion that the species remain dominantly allogamous:

(1) We investigated microscopically the gynostemia of several flowers of each of the three Azorean Platanthera species. In both P. pollostantha and P. micrantha the pollinia are located comparatively close to the stigmatic surface, the linear locular aperture appears relatively relaxed, and the pollinia tend to fragment into their component massulae (Figs. 11B, 11E, 11F, 13C and 14C). Each of these features could be invoked in support of the hypothesis of autogamy. However, in the case of the flowers that we examined, the massulae had rarely become attached to the stigma; it seems more likely that they had either become attached to a visiting insect or fallen past the stigma and out of the flower. And in flowers where massulae were found attached to the stigma, their location and aggregation appeared more consistent with deposition by an insect pollinator (Figs. 11E and 13C), comparable with their placement on the stigma of the unquestionably allogamous P. holmboei (Fig. 17A). Also, pollinaria are occasionally observed attached to vegetative organs of plants. Tyteca & Gathoye (2012) argued that this might constitute evidence of insect-mediated transport, though we suspect that the high winds often enjoyed by the islands may occasionally result in physical transfer of pollinaria between plants.

(2) The three species generate contrasting fragrances that are especially pronounced at night. We suspect that the same spectrum of volatiles is emitted by the flowers of each species but in different relative proportions. The basic odour, as produced by P. pollostantha, is musk-like with a spicy undertone, broadly resembling the strong scent famously produced by the Eurasian Fragrant Orchid, Gymnadenia conopsea s.s. In contrast, the fragrance emitted by P. micrantha resembles that of Freesia, having a sharper lemon-like undertone, and that emitted by P. azorica is weaker and less resinous to the human nose. Clearly, these fragrances merit quantitative biochemical analysis (cf. Plepys et al., 2002), but it seems unlikely that any of these scents would still be generously produced if they no longer fulfilled a pollinator attraction role (e.g., Roberts & Bateman, 2006).

(3) Similarly, the spurs of all three Azorean Platanthera species are reliably filled to between one third and one half of their length with sugar-rich nectar (e.g., Fig. 5D), representing a considerable investment of energy. It seems unlikely that natural selection would permit continued heavy investment in feeding pollinating insects if they were no longer required.

(4) Our genetic studies (Bateman et al., in press) identified at least some haplotypic diversity within about half of the populations of all three species, suggesting that gene flow is taking place among conspecific plants within those populations. In addition, we have evidence (albeit limited) that they lack the fixed heterozygosity that characterises autogamous species of some other Eurasian orchids, such as those within the genus Epipactis (Squirrell et al., 2002).

(5) We and other observers have detected evidence of hybridisation between P. pollostantha and P. micrantha, albeit considerably less frequent than anticipated. Our study revealed two hybrids out of 199 plants measured (1.0%). Other authors have reported an even lower frequency of hybrids; Rückbrodt & Rückbrodt (1994) recorded one hybrid plant out of ca 700 flowering plants examined, Delforge (2003) similarly identified one hybrid out of ca 850 plants examined (a frequency of ca 0.1% in both cases), and Tyteca & Gathoye (2012) reported two hybrids among 836 plants (0.2%). It seems unlikely that such cross-fertilisation events would take place without the assistance of an animate pollen vector.

Although the Platanthera species do not appear to be island autogams, they are nonetheless likely to lack intrinsic sterility barriers, thereby resembling many (arguably the majority) of orchid species, which tend to rely primarily on pre-zygotic barriers related to pollinator specificity (e.g., Scopece et al., 2007). In such self-compatible species, only one viable immigrant seed would, if fortunate enough to encounter a suitable mycorrhizal partner on arrival, be sufficient to establish the new island lineage (in this case, presumably forming a P. pollostantha-like population). The prevalence of such self-fertile lineages establishing themselves by emigrating from mainland populations to oceanic islands has been termed Baker’s Law (cf. Baker, 1955; Roberts & Bateman, 2006).

Phenological divergence

Even if pollinators are in principle available to cross-fertilise these orchids, another extrinsic mechanism that could contribute to reproductive isolation among these species is phenological divergence. For example, Delforge (2003) argued that flowering peaks a little later in P. micrantha than in P. pollostantha. Although peak flowering time was not estimated for all of our study populations (Table 1), we were able to compare flowering times for 18 populations of P. pollostantha, 10 populations of P. micrantha (six co-occurring with P. pollostantha) and the one known metapopulation of P. azorica (also co-occurring with P. pollostantha). These data provided two useful comparisons – firstly, of the relative phenological peaks of contrasting species pairs occurring in mixed populations, and secondly, of all conspecific populations against altitude.

Six populations in which P. pollostantha and P. micrantha co-existed were scored for phenology (Table 1). In five, the estimated peak flowering times ranged from precisely contemporaneous to micrantha preceding pollostantha by no more than one week. The notable exception was the two-week difference reported at Pico do Areeiro on São Jorge, located along the main west–east spine of the island just ca 1.3 km east of the sole metapopulation of P. azorica. Moreover, within that metapopulation, P. azorica was estimated to pre-flower co-occurring P. pollostantha by up to three weeks, suggesting that flowering of P. micrantha and especially P. azorica is comparatively accelerated along this ridge, despite the exposed habitats that it provides. Nonetheless, there still exists sufficient overlap of flowering between the three Platanthera species in the area to permit substantial gene flow between all possible species pairs.

When peak flowering was compared with altitude, the relatively narrow ranges of both parameters shown by P. micrantha meant that no statistically significant relationship was identified (Fig. 26). In contrast, linear regression suggested that flowering in P. pollostantha was delayed by an average of one day for each additional 14 m of altitude. Moreover, the earliest-flowering study population of P. pollostantha peaked 7–10 days before the other populations found at comparable altitudes, presumably because it occurred on the sunniest of the nine Azorean islands, Santa Maria.

Figure 26 Plot of peak flowering time versus altitude (m asl) for populations of Platanthera studied by us on the Azorean archipelago.

Overall, with the possible exception of precocious flowering in P. azorica on São Jorge, there is no evidence that phenological separation played a significant role in speciation of Platanthera on the Azores.

Functional morphology of the flower

Textbook evolutionary scenarios regarding pollinator specificity have been constructed on the back of observations made on members of the P. bifolia-chlorantha aggregate, particularly contrasting presumed attachment of viscidial discs to the compound eyes of moths by P. chlorantha (presumably also by P. holmboei and P. algeriensis, given their similar gynostemia) but to the proboscis of similar moths by P. bifolia, which reliably shows much shorter viscidial separation (e.g., Darwin, 1877; Nilsson, 1983; Hapeman & Inoue, 1997; Wood & Neiland, 2001; Maad & Nilsson, 2004; Little, Dieringer & Romano, 2005; Boberg et al., 2013: recently reviewed by Bateman, James & Rudall, 2012).

In this context, it is interesting to compare a plot of the distance separating the viscidia versus overall length of the pollinarium (Fig. 21B) with the overall dimensions of the gynostemium responsible for housing the pollinaria (Fig. 21A). All four metric characters wholly separate P. azorica from the two remaining Azorean species but show extensive overlap between P. micrantha and P. pollostantha, even though the latter has smaller mean values than the former for all four parameters. Nonetheless, the discrimination among the three species is greater in the two parameters that would routinely and directly influence the fit of the orchid’s male reproductive structures to the pollinating insect’s head, especially viscidial separation (Fig. 21B).

Our documentation of the details of gynostemium morphology in the six species illustrated using light micrographs (Fig. 11) and scanning electron micrographs (Figs. 12–17) encourages speculation regarding their likely interactions with pollinators. All three stigmatic lobes are evidently functional in all six species. The unusually small-flowered P. pollostantha and P. micrantha are striking in their relatively relaxed anther locules, which rapidly brown and desiccate when no longer in contact with massulae (e.g., Fig. 13C), and especially in the way that their short strap-like caudicles suspend the horizontal or slightly backward-oriented viscidia in front of the stigmatic surface (Figs. 11A, 11B, 13B and 14B–14D). In this regard, these species resemble neither the eye-fixing strategy of P. chlorantha, P. algeriensis, P. holmboei and P. azorica, nor the proboscis-fixing strategy of P. bifolia. The Pringle-shaped viscidial extension that characterises P. micrantha is especially intriguing. It is possible that the presentation of viscidia in these small-flowered species effects attachment of just one pollinarium to an insect, particularly if the insect approaches the spur entrance diagonally rather than perpendicularly (see below). Certainly, the viscidia have retained the requisite adhesive properties; inadvertently detached pollinaria proved to be a particular hazard when measuring the gynostemia.

Moving on to features most likely to be responsible for attracting pollinators prior to attachment of pollinaria, labellum length and spur length are, when plotted together, capable of discriminating all plants of all three species (Fig. 22B). The vast discontinuity that separates P. pollostantha from P. azorica is partially filled by P. micrantha, though it is not arithmetically intermediate between the other two species; rather, P. micrantha more closely resembles P. pollostantha in labellum length but more closely resembles P. azorica in spur length. The two putative hybrid plants between P. pollostantha and P. micrantha are placed between the parental clusters, albeit closer to the former than the latter. Plotting labellum length against labellum width also fully segregates the three species, though they are separated by narrower discontinuities (Fig. 22A). For at least these particular metric characters, the three species appear to efficiently apportion the available morphospace between them (cf. Bateman, 2001).

The many difficulties of both measuring and interpreting the functional significance of spur length in Platanthera (essentially a ‘moving target’ in both developmental and evolutionary terms) were discussed in detail by Bateman & Sexton (2008) and Bateman, James & Rudall (2012) and so need not be revisited in detail here. Presumably, the diagnostic disparities among the three Azorean species (Fig. 22B) have at least some influence over the spectra of pollinators that they entertain. All three species generate substantial quantities of nectar, though it is interesting that this output is achieved by the small-flowered species P. pollostantha and P. micrantha in the absence of the large papillae that feature on the spur interiors of all of their potential antecedents (Figs. 12–17) (Box et al., 2008; Bell et al., 2009).

The most striking feature of the labella of these plants is the contrast between the three species in the degree of reflexion or deflexion shown by the labellum. The comparatively large labellum of P. azorica is moderately to strongly reflexed in the mature flower (categories 3 and 4 in Fig. 20A). The labellum of P. pollostantha – similar to P. azorica in width but much shorter in length – is typically held near-vertically or more often projects slightly forward, albeit curving backwards slightly towards the apex. But the most striking labellum posture is the narrower and more flexible labellum of P. micrantha, which differs from the other two species in both projecting forward and curving upward (Figs. 5D and 14A). This unusual posture effectively prevents direct perpendicular access of insects to the gynostemium and presumably obliges them to access the nectar-rich spur obliquely, either to the left or to the right. Indeed, diagonal access to the spur could be facilitated by the unusual ‘letter box’ entrance that characterises both P. micrantha and P. pollostantha (e.g., Figs. 13C and 14C). Similar pollinator constraints have been observed in other Eurasian orchids characterised by small, green flowers, notably Herminium monorchis (Nilsson, 1979; Rudall, Perl & Bateman, 2013).

It is less clear whether the distinctive near-vertical (as viewed perpendicularly) position of the comparatively pale lateral sepals of P. micrantha – often causing the two spatulate sepals to overlap near the base – plays a role in pollinator attraction; however, it does allow reliable taxonomic distinction from the near-horizontal sepals of P. pollostantha and P. azorica (Fig. 20B). Lastly, a modest density of stomata was evident on the adaxial surfaces of not only the sepals (Figs. 12D and 13D) but also, more unusually among orchids, of the lateral petals. Only the labellum lacked stomata, though in compensation, putatively glandular cells were noted, increasing in frequency toward its apex (Figs. 13E and 15C). We assume that the relatively high density of chloroplasts in these Platanthera flowers permits significant photosynthetic and respirational activity, hence the presence of apparently functional stomata, while the glandular cells are most likely secretory. However, we suspect that the bulk of the strong floral fragrances that characterise the genus are generated by the unusually large auricles, which exhibit micromorphological features typical of osmophores.

Formal descriptions

All dimensions refer to fresh rather than dried plants. Variance in metric and meristic characters is given to two standard deviations, thereby in theory encompassing 96% of the plants measured. Variance in scalar characters is indicated by the following terms: usually = >80%, often = 51–80%, occasionally = 20–50%, rarely = <20%.

Quick key

1	Labellum > 7 mm, strongly reflexed;
lateral sepals > 7 mm; viscidia > 2 mm apart	P. azorica	
1*	Labellum < 7 mm, vertical/slightly reflexed;
lateral sepals < 7 mm; viscidia < 2 mm apart	2	
2	Labellum projecting forward and curving upward;
spur > 5 mm; lateral sepals nearer vertical than horizontal	P. micrantha	
2*	Labellum near-vertical, apex recurved;
spur < 5 mm; lateral sepals nearer horizontal than vertical	P. pollostantha	

Platanthera pollostantha R.M.Bateman & M.Moura, sp. nov.

Short-spurred Butterfly-orchid: Tubers broadly fusiform, narrowing to a single, long, fleshy apical root; a further 2–4 roots emerge horizontally from the base of the stem; old and new tubers separated by a substantial stolon 30–50% the length of the tuber. Stem 25 ± 17cm, 2.9 ± 2.0 mm in diameter. Sheathing leaves usually 2, largest 105 ± 73 mm × 31 ± 30 mm, broadly ovate/obovate, usually spreading and rarely with elongate petiole; bracteoidal leaves 4.1 ± 3.0, usually distributed fairly evenly along stem and grading into basal bracts. Inflorescence 77 ± 56 mm, 40 ± 36 flowers (19 ± 16 fls/cm). Basal bracts 27 ± 39 mm, floral bracts 10 ± 7 mm × 3.5 ± 2.0 mm, lanceolate; marginal cells rounded, 53 ± 24 µm in longitudinal diameter. Flowers uniformly schiele’s green to pea green (RHS 143C–144B in natural light, 149A in artificial flash); median sepal and lateral petals connivent over gynostemium. Labellum occasionally paler towards spur entrance, entire, 2.9 ± 1.4 × 2.1 ± 0.7 mm, elliptic-ovate, held vertically or more often projecting slightly backward but also usually curved gently backward. Spur 3.1 ± 0.8 mm long × 0.8 ± 0.5 mm in diameter at mouth, 0.8 ± 0.3 mm midway along its length, strongly down-curved; spur entrance strongly compressed vertically. Ovary 8.4 ± 3.6 mm. Lateral sepals oriented closer to horizontal than vertical, 3.4 ± 1.4 × 2.3 ± 0.7 mm. Lateral petals 2.3 ± 1.2 mm. Gynostemium 1.3 ± 0.5 mm long × 1.4 ± 0.5 mm wide; stigma immediately above spur entrance, a horizontally elongate oblong, at most 0.8 ± 0.6 mm wide; rostellum a subdued, near-horizontal ledge; auricles lateral to, and largely fused with, gynostemium, small, 0.5 ± 0.2 mm. Anther locules linked by a narrow, well-developed connective, locule aperture ± linear, relaxed; paired pollinaria 0.9 ± 0.4 mm, slightly to moderately convergent from viscidium to pollinium apex, viscidia separated by 0.8 ± 0.3 mm, apices of pollinaria by 0.5 ± 0.3 mm; viscidia pendent, angled inwards but not opposed, near-equidimensional, obscurely bipartite; caudicle near-linear, strap-like, much shorter than the pale yellow pollinium; pollinium bears few vertical rows of massulae. Fragrance strong, almost resinous, of musk and spice. ITS1 includes the motif TTCAACTACA; ITS2 includes the motif CTCAATCGTT.

Distribution: Endemic to the Azores, occurring on all islands; frequency differs between islands according to the areal extent of, and degree of anthropogenic disturbance suffered by, land above 400 m asl (hence, the species is locally frequent on, for example, Pico and São Jorge).

Habitat: Most frequent in laurisilva scrub, rough grassland above lavas and alpine grassland; also found in rough pastures, oakwoods and Cryptomeria plantations; (250–)400–1000(–1300) m asl.

Holotype: TUB008187, KCF Hochstetter, 1838 (no original label retained); later annotated “Platanthera micrantha (Hochst.) Schltr. (Abbildungsvorlage zu: Seubert, Flora Azorica Tab. V, Fig. 1), Ggf. Epitypus, rev. Martin Engelhardt, 22. Juli 1993” (here shown as Figs. 28A and 28B). Specimen first designated here, despite its considerable age.

Illustrations:Figs. 4, 11A, 11E, 13, 28A and 28B.

Etymology: Novel epithet derived from the Greek pollostos (smallest, least) and anthos (flower), reflecting the fact that the flowers of this species are even smaller than those of P. micrantha.

Platanthera micrantha (Hochstetter ex Seubert) Schlechter, Repert. Spec. Nov. Regni Veg. 16: 378 (1920), emend R.M.Bateman & M. Moura

Basionym: Habenaria micrantha Hochstetter ex Seubert, Fl. Azor.: 25 (1844)

Narrow-lipped Butterfly-orchid: Tubers broadly fusiform, narrowing to a single, long, fleshy apical root; a further 2–4 roots emerge horizontally from the base of the stem; old and new tubers separated by a substantial stolon 30–50% the length of the tuber. Stem 32 ± 18cm, 3.7 ± 2.4 mm in diameter. Sheathing leaves often 2, occasionally 3, largest 125 ± 73 mm × 54 ± 31 mm, broadly ovate/obovate, spreading and lacking an elongate petiole; bracteoidal leaves 3.3 ± 2.6, usually distributed fairly evenly along stem and grading into basal bracts. Inflorescence 109 ± 80 mm, 60 ± 60 flowers (18 ± 16 fls/cm). Basal bracts 21 ± 27 mm, floral bracts 13 ± 6 mm × 4.9 ± 2.0 mm, lanceolate; marginal cells rounded, 47 ± 16 µm in longitudinal diameter. Flowers pea green to agathia green (RHS 142D–149D in natural light, 144D in artificial flash); median sepal ± horizontal and lateral petals torsioned inwards to arch over gynostemium. Labellum often paler towards spur entrance, entire, 4.6 ± 1.6 × 1.6 ± 0.6 mm, linear-lanceolate, projecting clearly forwards but also curved moderately to strongly forwards, obscuring spur entrance. Spur 7.3 ± 1.9 mm long × 0.8 ± 0.5 mm in diameter at mouth, 0.9 ± 0.3 mm midway along its length, usually strongly down-curved; spur entrance strongly compressed vertically. Ovary 11.9 ± 3.5 mm. Lateral sepals oriented closer to vertical than horizontal, 4.9 ± 1.4 × 2.8 ± 0.9 mm. Lateral petals 3.2 ± 0.9 mm. Gynostemium 1.5 ± 0.5 mm long × 1.5 ± 0.5 mm wide; stigma immediately above spur entrance, a horizontally elongate oblong, at most 1.2 ± 0.4 mm wide; rostellum a subdued, near-horizontal ledge; auricles lateral to, and largely fused with, remainder of gynostemium, small to almost absent, 0.5 ± 0.4 mm. Anther locules linked by a narrow, well-developed connective, locule aperture ± linear, relaxed; paired pollinaria 1.2 ± 0.3 mm, slightly to moderately convergent from viscidium to pollinium apex, viscidia separated by 1.0 ± 0.4 mm, apices of pollinaria by 0.6 ± 0.3 mm; viscidia pendent, angled inwards but not opposed, strongly ellipsoidal and concave, clearly bipartite; caudicle near-linear, strap-like, much shorter than the pale yellow pollinium; pollinium bears few vertical rows of massulae. Fragrance combines Freesia and Citrus. ITS1 includes the motif TTCAACTACA; ITS2 includes the motif CTCAATTGTT.

Distribution: Endemic to the Azores, possibly still occurring on all islands except Graciosa (present status on Santa Maria and Terceira uncertain); frequency differs between islands according to the areal extent of, and degree of anthropogenic disturbance suffered by, land above 300 m asl (hence, the species is scattered as mostly small populations on, for example, Pico and São Jorge).

Habitat: Largely confined to, and a good indicator of, laurisilva scrub and adjacent rough grassland; 300–900(–1100) m asl.

Holotype: TUB010453, KCF Hochstetter, 1838 (original label details: “Gymnadenia micrantha Hochst., Habenaria ejusd. olim., n./p., in montolis[?] insularum Azoricum, Majo 1838, altit. 1500–2000’, flores viriscentes, leg. Car. Hochstetter”) (here shown as Figs. 28C and 28D).

Illustrations:Figs. 5, 11B, 11F, 14, 28C and 28D.

Platanthera azorica Schlechter, Repert. Spec. Nov. Regni Veg. 16: 378 (1920), emend R.M.Bateman & M.Moura

Synonym: Habenaria longebracteata Hochstetter ex Seubert, Fl. Azor.: 25 (1844)

Hochstetter’s Butterfly-orchid: Tubers broadly fusiform, narrowing to a single, long, fleshy apical root; a further 2–4 roots emerge horizontally from the base of the stem; old and new tubers separated by a substantial stolon 30–50% the length of the tuber. Stem 20 ± 10 cm, 3.8 ± 2.0 mm in diameter. Sheathing leaves usually 2, occasionally 1, largest 111 ± 64 mm × 40 ± 24 mm, broadly ovate/obovate, spreading and lacking an elongate petiole; bracteoidal leaves 1.8 ± 1.4, usually distributed fairly evenly along stem and grading into basal bracts. Inflorescence 85 ± 46 mm, 18± 11 flowers (4.7 ± 2.8 fls/cm). Basal bracts 28 ± 17 mm, floral bracts 18 ± 7 mm × 5.4 ± 1.8 mm, lanceolate; marginal cells rounded, 51 ± 16 µm in longitudinal diameter. Flowers uniformly agathia green (RHS 142B, C in natural light, 140A–144D in artificial flash); median sepal forms an operculum over the lateral petals, which project directly forward, together forming an ‘awning’ above the gynostemium. Labellum occasionally paler toward spur entrance, entire, 8.3 ± 2.0 × 2.4 ± 0.5 mm, linear-lanceolate, recurved moderately to strongly backward and also curved gently to moderately backward. Spur 9.5 ± 1.8 mm long × 1.4 ± 1.1 mm in diameter at mouth, 1.1 ± 0.5 mm midway along its length, strongly or occasionally moderately down-curved; spur entrance ± equidimensional rather than compressed vertically. Ovary 14.0 ± 2.8 mm. Lateral sepals oriented closer to horizontal than vertical, 8.2 ± 1.8 × 3.6 ± 0.9 mm. Lateral petals 5.7 ± 1.7 mm. Gynostemium 3.2 ± 1.0 mm long × 3.5 ± 0.8 mm wide; stigma immediately above spur entrance, a horizontally elongate crescent, 2.7 ± 1.0 mm wide; rostellum a subdued, elongate crescent; auricles lateral to, and largely fused with, remainder of gynostemium, comparatively large, 0.8 ± 0.9 mm. Anther locules linked by a wide, well-developed connective, robust, locule aperture sigmoid, relatively taut; paired pollinaria 2.5 ± 1.1 mm, slightly to moderately convergent from viscidium to pollinium apex, viscidia separated by 3.1 ± 0.7 mm, apices of pollinaria by 2.2 ± 0.6 mm; viscidia ± opposed, orbicular, obscurely bipartite; caudicle proximally geniculate, terete, ± equalling the pale yellow pollinium; pollinium bears several vertical rows of massulae. Fragrance comparatively subtle, of musk and spice. ITS1 includes the motif TTCAACTACA; ITS2 includes the motif CTCAATCGTT (shared with P. pollostantha).

Distribution: Endemic to the Azores, found in recent years only on one volcanigenic ridge on São Jorge.

Habitat: Alpine grassland and open, dwarfed laurisilva scrub at ca 950–1000 m asl.

Holotype: TUB010453, KCF Hochstetter, 1838: (original label details: 114. Gymnadenia longebracteata Hochst., Habenaria ejusd. olim. mscpt., locis graminolis in regionibus elatioribus insularum Azoricum, Junio/Julio 1838, legit Carolus Hochstetter, flores viriscentes, tuberibus oblong-rotunda” plus, in a different (and seriously erroneous) hand, “Coeloglossum viride”. Subsequently annotated “Platanthera azorica Schltr. (Abbildungsvorlage zu: Seubert, Flora Azorica Tab. V, Fig. 2), Ggf. Epitypus, rev. Martin Engelhardt, 10. Juni 1995)” (here shown as Figs. 28E and 28F).

Illustrations:Figs. 6, 11C, 15, 28E and 28F.

An exceptionally convoluted taxonomic history

The problem

The two widely recognised Azorean species of Platanthera were first named (under the genus Habenaria) as elements of a floristic list by Seubert & Hochstetter (1843). However, they were not formally described or illustrated until the following year, when Seubert (1844) published the first explicit flora of the islands. Seubert’s (1844, 25) formal Latin descriptions of these species have been universally acknowledged to be woefully inadequate; for example, Seubert’s contemporary Watson (1870, 114) commented (waspishly but perceptively) that “Dr. Seubert was placed under the inconvenience and great disadvantage of writing the Flora of a country which he had not seen. Thus his work is truly more a botanical account of dried specimens from the Azores Isles than a proper Flora of those isles; and perhaps it would have been better had he even more strictly limited himself to such an account, avoiding guesses that might prove [to be] only erroneous records”. This point is well-illustrated by a comment made by Seubert (1844, 25; see Table 4) himself and pertaining to the butterfly-orchids: “Because the flowers are dried, we are unable to judge whether [or not] they belong to the Gymnadeniae” – in other words, the herbarium specimens in question could not even be assigned to the appropriate subtribe, let alone the correct genus, due to the inability to access adequately diagnostic characters.

Figure 27 Original line drawings depicting the holotypes of P. micrantha (left: labelled Gymnadenia/Habenaria micrantha) and P. azorica (right: labelled Gymnadenia/Habenaria longebracteata).

Reproduced from plate V of Seubert (1844).

Figure 28 Images of whole plants and magnified images of the best-preserved flowers (circled) of the holotypes of the three Azorean Platanthera species.

All specimens were collected in 1838 by Karl Hochstetter and are currently held at the University of Tübingen. (A, B) P. pollostantha. (C, D) P. micrantha. (E, F) P. azorica. Compare (C) and (E) with the original line drawings shown as Fig. 27. Scale bar for (B, C, E) = 10 mm. Images: R Bateman.

These technical criticisms certainly apply to Seubert’s account of the two ‘Habenaria’ species. The descriptions (English translations of the original Latin are presented in Table 4) are even more sparing than is required by their origin in herbarium specimens, the phrasing is undesirably vague and qualitative, and there is no consistency between the two descriptions in either the choice of characters or the terms used to describe them. Most of the few characters listed by Seubert adequately describe all three Azorean species of Platanthera, and some are sufficiently generalised to describe all European species of the genus. Also, Seubert’s (1844) accounts of preferences in habitat type and altitudinal range of Azorean plants are notoriously unreliable (Watson, 1870).

Table 4 The original descriptions of P. pollostantha (as Habenaria micrantha) and P. micrantha (as Habenaria longebracteata).

Reproduced from the flora of Moritz Seubert (1844, 25), which was based on herbarium specimens collected in the Azores in 1838 by Karl Hochstetter and passed on to his father Ferdinand Hochstetter. They could not be poorer; characters are few and imprecise (translation from the original Latin and German text).

155. Habenaria micrantha (Hochstetter msc.): labellum entire, oblong or linear-oblong, blunt, equalling lateral sepals; spur club-shaped, curved, half the length of the ovary; bracts three- or five-veined, exceeding the lower flowers; tubers undivided. – Plate V, Fig. 1 + 1a (flower viewed from side).	
Habitat: In mountains, alt. 1500′–2000′ (for example, rarely on Pico).	
Plant terrestrial/herbaceous. Lowest leaves reduced to sheaths, two intermediate leaves large, elliptical, upper leaves gradually thin into floral bracts. Flowers small, green, numerous in a dense spike.	
This and the following species apparently have great affinity with …American Habenarias (Platanthera Lindl.). Because the flowers are dried, we are unable to judge whether they belong to the Gymnadeniae.	
156. Habenaria longebracteata [ = azorica Schltr.] (Hochstetter msc.): labellum entire, linear, apex blunt; lateral sepals spreading, spur filiform, shorter than the ovary; bracts multi-veined, exceeding flowers; tubers undivided. – Plate V, Fig. 2 + 2a (flower viewed face-on, at top of plate).	
Habitat: In flat grassy areas (Coll. No. 114).	
Tubers oblong, leaves paired and flowers green, as in Platanthera bifolia and with a similar[ly small?] number of flowers. Congeneric with American [?species] …it is easily distinguished from, e.g., Habenaria bracteata R.Br. [ = Dactylorhiza/Coeloglossum viridis in modern classifications] by the much larger, lax spike.	

Readers of Seubert’s descriptions are left grasping at straws when seeking linguistic clues regarding the true identity of the two orchids in question. The only even semi-quantitative character presented is “spur …half the length of the ovary” given for ‘H. micrantha’. Unfortunately, a glance at Fig. 23A immediately demonstrates that spurs of H. pollostantha (i.e., ‘H. micrantha’) average one third the length of the ovary (but with an exceptionally broad spread of data for ovary length) and those of the other two Azorean Platantheras average two-thirds the length of the ovary. Moreover, ovary length incurs a far greater coefficient of variation than spur length in P. pollostantha (Table 3), and linear regressions of those two variables for each species do not pass through the origin of the graph. In truth, the only useful comments in Seubert’s accounts are found toward the end of the description of ‘H. longebracteata’, where the species is compared with P. bifolia as sharing a lax inflorescence containing few flowers – a description applicable only to P. azorica (sensu the present study) among the three Azorean species. Thus, it is hardly surprising that the identity, circumscription and taxonomic affinity of these species have been much debated during the 170 years elapsed since publication of Seubert’s flora (cf. Trelease, 1897; Schlechter, 1920; Schlechter, 1923; Rückbrodt & Rückbrodt, 1994; Delforge, 2003). Several authors commented on the indisputably poor fit between Seubert’s (1844) written descriptions and the associated plate of two large and two small drawings, which fortunately are of higher quality (Fig. 27).

Our breakthrough in taxonomic understanding of these plants occurred through a combination of Moura’s serendipitous field discovery on São Jorge on 23rd June 2011 and Bateman’s subsequent request to borrow the Hochstetter holotypes of ‘Platanthera micrantha’ and ‘P. azorica’ from the herbarium at Tübingen, Germany; curator Cornelia Dilger sent him three herbarium sheets, rather than just the two holotypes. Two of these sheets unequivocally bore the two specimens that were illustrated by Seubert (1844) (Figs. 28C–28F), but a third specimen – evidently part of the same 1838 collection made by Hochstetter – was also sent, bearing a much later annotation of “Platanthera micrantha …Epitypus” added by Martin Engelhardt on 22nd July 1993 (Figs. 28A and 28B). Even the most cursory examination of these specimens was sufficient to show that the holotype clearly attributed by Seubert (1844) to ‘Habenaria micrantha’ is in fact acceptably representative of the much rarer species that has been described by all subsequent authors as Platanthera azorica ( = ‘H. longebracteata’ as originally formally described by Seubert). Even more remarkably, the holotype of P. azorica accurately exemplifies not the species referred to by all subsequent authors as P. azorica but rather what we had previously believed to be a new species of Platanthera discovered by us in 2011 on the ‘spine’ of São Jorge and provisionally named by us as P. ‘adelosa’. Instead, we must now accept that this exceptionally rare species was originally found by Hochstetter in 1838 but not seen again (or at least not again recognised as being a distinct taxonomic entity) until 2011. And most remarkably of all, the third Hochstetter specimen provided to us by Tübingen (Figs. 28A and 28B), which was not illustrated or mentioned by Seubert (1844), ably represents the most widespread Azorean Platanthera species – that previously ascribed to P. micrantha. It seems most likely that Hochstetter (filius) recognised all three species during his 1838 collecting tour of six of the nine Azorean islands, but that his taxonomic intentions were not communicated with sufficient clarity, either to his father or to Seubert, whose brief and highly ambiguous descriptions considered only two of Hochstetter’s three excellent specimens, each of which presumably intended as a holotype.

Even setting aside the ensuing 170 years of chronic taxonomic confusion, the consequences of this error are profound. Stated simply, the relationship between epithets and types becomes shifted sideways by one step, in a process resembling a ‘reading frame shift’ when translating DNA codons into amino acids. The most frequent of the three species, until now widely known as P. micrantha, does not presently have a legitimate Linnean epithet or a published or figured type. The less frequent (but widely accepted) species commonly known as P. azorica has as its holotype the specimen that for the last 170 years has been universally considered to represent (albeit badly) P. micrantha. And the exceptionally rare “new” species from São Jorge has as its holotype the specimen that for the last 170 years has been universally considered to represent P. azorica!

The (regrettable) solution

Once we had finally reached this distressing conclusion regarding the true nature of the holotypes, our initial intention was to attempt nomenclatural conservation of the epithets micrantha and azorica so that they could continue to be applied to the species that have reliably borne these epithets for the last 170 years. Such a solution would, in our view, be justified by the familiarity of these epithets, the inappropriateness of their application to the legally correct species (‘micrantha’ is no longer the smallest-flowered species on the islands, and ‘azorica’ is presently known from only one portion of one Azorean island, rather than epitomising the entire archipelago), and the long-term confusion that will inevitably be caused by the ‘reverse frame-shift’ in nomenclature needed to at last link the valid epithets to the relevant holotypes.

Indeed, some rules of the International Code of Nomenclature for Algae, Fungi and Plants (McNeill et al., 2012) encourage such a bid for conservation:

“14.1. In order to avoid disadvantageous nomenclatural changes entailed by the strict application of the rules, and especially of the principle of priority …, this Code provides …lists of names of families, genera, and species that are conserved.

14.2. Conservation aims at retention of those names that best serve stability of nomenclature”. And:

“14.9. A name may be conserved with a different type from that designated by the author or determined by application of the Code”.

However, other rules appear mutually contradictory:

“57.1. A name that has been widely and persistently used for a taxon or taxa not including its type is not to be used in a sense that conflicts with current usage”. But:

“51.1. A legitimate name must not be rejected merely because it, or its epithet, is inappropriate or disagreeable, or because another is preferable or better known, or because it has lost its original meaning”.

In addition, it has been the experience of one of us (cf. Bateman et al., 2010; Brummitt, 2011) that, in practice, the welcome pragmatism encapsulated in Article 14.2 cannot be relied upon to override determined application of the often unwelcome core principle of priority stated in Article 11.4:

“11.4. For any taxon below the rank of genus, the correct name is the combination of the final epithet of the earliest legitimate name of the taxon in the same rank, with the correct name of the genus or species to which it is assigned”.

Hence, strict implementation of the Code means that the choice of valid Linnean epithet must be dictated by the nature of the original holotype, which always takes precedence over the content of the formal diagnosis in the protologue. Thus:

“9.1. A holotype of a name of a species or infraspecific taxon is the one specimen or illustration used by the author, or designated by the author as the nomenclatural type. As long as the holotype is extant, it fixes the application of the name concerned”.

Admittedly:

“38.1. In order to be validly published, a name of a new taxon must be accompanied by a description or diagnosis of the taxon”.

However, the accuracy and content of any formal diagnosis lies outside the jurisdiction of the Code; regrettably, there are no required minimum standards for taxonomic description (cf. Bateman, 2011). And, in any case, Article 38.1 does not apply to the Azorean Platantheras, as their protologues date from 1844:

“38.8. The name of a new species or infraspecific taxon published before 1 January 1908 may be validly published even if only accompanied by an illustration with analysis”.

All of these rules ultimately point towards the crucial relationship between epithet and type specimen – as determined in this case by the crucial intermediaries represented by the four etchings of two holotypes (reproduced here as Fig. 27) that were published as part of the protologue by Seubert (1844). Character-based evidence linking the formal Latin description of ‘Habenaria longebracteata’ (later Platanthera azorica) to the relevant drawings and holotype is circumstantial, and that linking the description of ‘Habenaria microphylla’ to its line drawing and holotype is non-existent. However, this fact is deemed irrelevant, as there is no such ambiguity in the references to those drawings that terminate the formal diagnoses, and thus firmly link the epithets ‘microphylla’ and ‘azorica’ to Figs. V.1 + 1a and V.2 + 2a, respectively. These figures then lead unambiguously to the two illustrated holotypes still held in the Tübingen herbarium.

Thus, the biological species formerly known as P. azorica of necessity receives the pre-existing name P. micrantha, the ostensibly ‘new’ biological species from São Jorge receives the pre-existing name P. azorica, and the most widespread of these biological species, formerly known as P. micrantha, is here given the novel epithet P. pollostantha (“smallest flowered”) – an epithet chosen by us to emphasise that it is this species, rather than P. micrantha, that possesses the smallest flowers among the three Azorean butterfly-orchids. We can finally close this 170-year-old circle using the ‘Third Specimen’ (an individual analogous to Graham Greene’s ‘Third Man’ – for long hidden from view but crucial to solving the entire mystery). Collected in 1838 by Karl Hochstetter, lodged in Tübingen and later insightfully annotated “Epitypus” by Martin Engelhardt (Figs. 28A and 28B), this specimen will serve as a taxonomically satisfactory and historically contemporaneous holotype for the never previously validly named P. pollostantha.

Pertinent postscript

The obscure details surrounding the field acquisition of the three Hochstetter holotypes proved to be of greater practical relevance than we originally envisaged. They were collected not by the comparatively well-known botanist Christian Ferdinand (CFF) Hochstetter (1787–1860) but rather by his son, Karl (KCF/CCF) Hochstetter (1818–1880). For several months in the summer of 1838, Hochstetter filius toured three Azorean islands – São Miguel, Terceira and Faial – with botanist Heinrich Guthnick and mineralogist Rudolph Gygax, before setting out alone to collect on a further three islands – Pico, Flores and Corvo (Jorge et al., 2011). Thus, Karl Hochstetter did not visit São Jorge, therefore he almost certainly collected the holotype of the “new” P. azorica on a different island. Ergo, our Pico da Esperança locality on São Jorge is not in fact a rediscovery of the species but rather a new discovery on a new island, raising the distinct possibility that this exceptionally rare orchid remains to be (re)found on at least one further Azorean island.

Serious constraints on herbarium-based taxonomy

Rückbrodt & Rückbrodt (1994) justly questioned the ability of Seubert to infer from Hochstetter’s herbarium specimens (Fig. 28) diagnostic characters such as the positions of the (lateral) sepals and the position and curvature of the labellum. However, our data show that loss of information in mounted herbarium specimens extends well beyond those characters that depend on three-dimensionality (or, indeed, on colour). Importantly, the sizes and shapes of organs are also radically altered.

One advantage of gathering our large body of morphometric data and carefully examining Hochstetter’s holotypes is that we can assess retrospectively whether each holotype is acceptably representative of the species for which, by definition, it is the archetype. The type specimens have been heavily criticised by many past authors (e.g., Watson, 1870; Trelease, 1897; Schlechter, 1920; Rückbrodt & Rückbrodt, 1994; Delforge, 2003), but this is not surprising, since each of these authors was attempting to connect the holotypes to the wrong species! Morphometric measurements for those characters that could realistically be measured in the three holotypes are given in Table 3 and explicitly distinguished wherever possible in Figs. 18–24.

Bateman & Rudall (2006) reported that flowers of Dactylorhiza fuchsii mounted on double-sided adhesive tape prior to morphometric assessment shrank by an average of 0.7% for four parameters and by only 2% for even the most divergent of those parameters. In contrast, dimensions of the flowers of the three Azorean holotypes – which like almost all herbarium specimens have not been glued to shrinkage-resistant backing – deviated considerably from the taxon mean values derived from our morphometric survey of in situ plants. For the 12 metric characters that could be measured in the flowers of the holotypes, they were on average smaller than the taxon mean by 18.3 ± 10.9% for P. pollostantha, 10.3 ± 5.1% for P. micrantha and 20.6 ± 13.4% for P. azorica (Table 3). Considering spur length as an example, shrinkage in the holotypes is estimated as 23% for P. pollostantha, 11% for P. micrantha and 17% for P. azorica (Fig. 22B).

Even more problematically, in the majority of floral organs the degree of shrinkage is evidently strongly non-allometric. For example, the labella of the three holotypes deviate from mean values for fresh flowers by an average of 14% shrinkage in length but by 33% shrinkage in width (Fig. 22A). Similarly, the lateral sepals show an average of 6% shrinkage in length (Fig. 24A) but 22% shrinkage in width (data not shown). Such strongly non-uniform shrinkage substantially alters the observer’s perceptions of not just the sizes but also the shapes of the affected organs. It is particularly striking that, when the types were included in the principal coordinates analysis of floral characters only for the three Azorean species (Fig. 18B), all three type specimens lay outside the clusters formed by conspecific living plants. Clearly, a taxonomic description based on herbarium material would constitute a seriously misleading guide to identification if subsequently applied to living plants in the field. In other words, the field-based morphometric approach to taxonomy advocated here yielded species circumscriptions and diagnostic characters that are far more reliable than any generated in herbaria (e.g., Bateman, 2012). Any attempt to compare herbarium specimens with field plants would benefit from assessments of shrinkage in specimens measured in the field that have subsequently been incorporated into the relevant herbarium.

Fortunately, if appropriate adjustments are made for herbarium shrinkage, each of the three holotypes appears to be acceptably representative of its source species in most characteristics. However, there are still some substantial deviations from the taxon mean values: the holotype of P. pollostantha has basal leaves with much greater width/length ratios than are typical of the species, whereas the converse is true of the holotype of P. azorica (Fig. 23B). The holotype of P. pollostantha also has unusually short basal bracts and that of P. micrantha has unusually long lateral petals, as well as a tendency towards fasciation at the apex of the inflorescence (a frequent feature among the larger individuals of this species). Lastly, the holotype of P. azorica has an unusually dense inflorescence (6.5 fls/cm, versus a taxon mean of 4.7 fls/cm: Table 3).

Factors influencing species distributions

Distribution among islands

The original descriptions of P. pollostantha (as Habenaria micrantha) and P. micrantha (as H. longebracteata = azorica) by Seubert (1844) gave no indication of the distributions of either species among the islands. Within a quarter-century, Watson (1870) felt able to report what we assume to be P. pollostantha from five of the nine islands (Flores, Faial, Pico, São Miguel, Santa Maria) and P. micrantha (as H. longebracteata) from three islands (Flores, São Miguel, Santa Maria). More recently, Aguiar, Fernández Prieto & Dias (2006) (see also Borges et al., 2010) added São Jorge to the previous list of five islands yielding P. pollostantha, whereas Schäfer (2005) listed this species from all Azorean islands other than Graciosa. Our own survey (Table 1, Fig. 1) located P. pollostantha on all islands except the lowest, Graciosa, and in 2011, Graciosa Island Park staff found a population there – a fact that is confirmed by the relevant map in the Azores Bioportal (Silva, 2013) and supports previous assertions by Frey & Pickering (1975) and Sjögren (1984) that this species occurs on every island.

In contrast, Delforge (2006) reported P. micrantha from only five islands (Flores, Pico, São Jorge, São Miguel, Santa Maria), echoing previous assessments by Palhinha (1966), Sjögren (1973), Frey & Pickering (1975) and Buttler (1991), while Schäfer (2005), Aguiar, Fernández Prieto & Dias (2006) and Tyteca & Gathoye (2012) dropped Santa Maria from this list. Indeed, despite these earlier records, we were unable to re-find P. micrantha on Santa Maria, nor is the species recorded for that island in the Azores Bioportal. In partial compensation for these possible losses, two small populations of P. micrantha were found on Corvo in 2007 by Pereira et al. (2007, their Fig. 2h). Also, Schäfer (2002) reported the occurrence of P. micrantha on Faial; one population was found in the Faial caldera by Mark Carine in 2008 and a further small population was detected nearby by Rudall and Bateman in 2011. However, neither our field expeditions nor any others contributing to the Azores Bioportal and its successor, Atlantis 3.1, have located P. micrantha on either Graciosa or Terceira.

Altitude

The nine Azorean islands range in maximum altitude from 402 m on Graciosa to a monumental 2351 m on Pico, leading to several previous comments of varying degrees of accuracy regarding the altitudinal preferences of these orchid species (cf. Seubert, 1844; Sjögren, 1973; Frey & Pickering, 1975; Rückbrodt & Rückbrodt, 1994; Delforge, 2006). Table 5 summarises altitudinal data for 140 Platanthera populations, derived from the present study plus those of Rückbrodt & Rückbrodt (1994) and Delforge (2003). Populations of P. pollostantha range from an exceptional 240 m on São Miguel (Delforge, 2003) to 1330 m on Pico (Rückbrodt & Rückbrodt, 1994), whereas those of P. micrantha show a slightly narrower range, extending from ca 330 m on Flores, Corvo and São Jorge to 1110 m on Pico. Mean altitudes per island range from 368 m for P. micrantha on Corvo to 815 m for P. pollostantha on Pico and Faial (Table 5).

Table 5 Altitudinal ranges (m asl) of Platanthera pollostantha and P. micrantha on each Azorean island.

Ranges were determined by combining data from the present study with locality lists appended by Rückbrodt & Rückbrodt (1994) and Delforge (2003).

Island	Maximum
altitude	pollostantha					micrantha					
		Lowest	Highest	Mean	SSD	n	Lowest	Highest	Mean	SSD	n	
Flores	886	594	683	639	NA	2	325	594	460	NA	2	
Corvo	718	414	600	522	96	3	321	414	368	NA	2	
Faial	1043	780	850	815	NA	2	731	780	756	NA	2	
Pico	2351	520	1330	815	161	36	520	1110	744	147	16	
São Jorge	1083	420	1000	684	180	16	330	885	630	174	11	
Graciosa	402	ND				0	ND				0	
Terceira	1021	475	970	677	228	6	ND				0	
São Miguel	1105	240	810	598	125	31	500	820	683	94	8	
Santa Maria	587	393	474	445	45	3	ND				0	
Total						99					41	
Notes.

SSD sample standard deviation

ND no data

NA not applicable

It is perhaps more instructive to compare mean altitudes for the two species on each island, in order to allow for topographical differences. Platanthera pollostantha appears to prefer somewhat higher altitudes than P. micrantha on the western isles of Flores and Corvo, though available data are undesirably sparse. The much more heavily populated datasets for Pico, São Jorge and São Miguel indicate at most only modest differences between the species in altitudinal preference; P. pollostantha averages slightly higher altitudes than P. micrantha on Pico and São Jorge, reflecting its greater ability to stretch upward into the alpine zone above 900 m asl, whereas the converse relationship between the two species is evident on São Miguel (Table 5). Most populations of both species occur between 400 m and 900 m asl, perhaps explaining why Graciosa (peaking at a mere 402 m asl, and consequently highly cultivated) has failed to provide conducive habitats. Lastly, current knowledge suggests that P. azorica essentially forms just a single metapopulation centred on Pico da Esperança, the highest point on São Jorge at 1083 m asl. Here, the plants appear to be confined to a narrow zone between about 950 and 1000 m asl. It is uncertain whether this species has in the past extended further down the slopes of the volcanic ‘spine’ of São Jorge, though the continued presence of extensive semi-natural dwarfed laurisilva forests on the northeast slope of the ridge (i.e., below Pico da Areeiro) suggests that the species has always been rare on the island and confined to the ridge crest. Thus, with the possible exception of P. azorica, there is at best limited evidence that altitudinal segregation could have contributed to speciation of the Azorean Platantheras.

The apparent preference of P. pollostantha and P. micrantha for intermediate altitudes could be interpreted as evidence of the “mid-domain effect” sensu Colwell & Lees (2000). This hypothesis states that occupying intermediate altitudes on a topographically variable island increases the longevity of a species, because given continuity of habitat, the species can respond to changes in the local climate by migrating either upward or downward. In contrast, a low-level coastal plant can only migrate upward, and a mountain-top species can only move downward; in both cases, this unidirectionality presumably doubles their theoretical risk of extinction. Thus, global warming must be considered a particular threat to P. azorica, as this species appears to be confined to the top of a volcanic ridge on a single island (cf. Roberts & Bateman, 2006).

Preferred habitats

According to Aguiar, Fernández Prieto & Dias (2006), the preferred phytosociological association of P. pollostantha is Tolpido azoricae–Holcetea rigidi, whereas that of P. micrantha was given as Festucium jubatae. Schäfer (2005) reported the topographic location of P. pollostantha as “volcanic craters [and] juniper rainforest”, and that of P. micrantha as “volcanic craters [and] steep slopes”.

Our observations suggest that P. micrantha is more tightly tied to the less shaded categories of laurisilva forest (associated with shrubs such as Vaccinium cylindraceum, Viburnum treleasei, Ilex perado, Laurus azorica, Juniperus brevifolia) and with volcanic features such as calderas, parasitic cones and lava tunnels (Fig. 2). These habitats also reliably support P. pollostantha, but it is also capable of occupying both denser forest (including non-native Cryptomeria plantations) and exposed habitats such as high-altitude Erica–Daboecia scrubland, Calluna heathland and peat bogs (Fig. 3). This species is even occasionally epiphytic in dense patches of moss, especially on ancient junipers. In his reassessment of plant communities of the Azores, Dias (1996) described P. pollostantha as being characteristic of mesic laurifoliate forest and Juniperus forest on peatlands, and as occasional in hyper-humid laurifoliate forest and local stress shrublands. Give that only one metapopulation of P. azorica is presently known, it is difficult to generalise regarding its habitat preference. It is found in high-altitude laurisilva scrub that is so sparse and so dwarfed that it arguably qualifies as grassland (Fig. 3A). Woody species present are typical of laurisilva, but notable prominent herbs include Ranunculus cortusifolius and Tolpis azorica. More generally, we would be especially interested in learning how much laurisilva formerly occurred at lower altitudes on the islands. We suspect that populations of P. pollstantha and possibly P. micrantha were once more frequent below 400 m asl but were largely eradicated through anthropogenic habitat destruction.

Soil pH values on the Azores are characteristically fairly uniform and moderately to strongly acidic, reflecting the geochemistry of the ubiquitous soil-forming volcanic rocks (Table 1). Field measurement of soil pH at the single locality for P. azorica on São Jorge yielded a relatively high (in Azorean terms) value of 5.9, whereas pH measurements from two populations of P. micrantha (both co-occurring with P. pollostantha on São Jorge and Pico) were more acidic at 5.2 and 4.1, respectively. Measurements from eight populations of P. pollostantha spanning six islands produced pH values that ranged from 4.1 to 6.1 and averaged 5.4.

Returning briefly to the ecology of P. micrantha, we noted that this species is usually (perhaps always) found in close association with Vaccinium cylindraceum, a key element of high-quality laurisilva on the Azores (Pereira, 2008). Excavation around the respective roots of these two species suggested that they may share a single mycorrhizal associate (Bateman et al., in press), a hypothesis that now requires carefully targeted testing. Certainly, we regard P. micrantha as an especially valuable indicator of high-quality semi-natural vegetation on the archipelago.

Conservation value

Estimating population sizes and hybrid frequencies

Previous authors have reported that P. pollostantha is far more frequent than P. micrantha, and that hybrids between them are rare: numbers of flowering plants reported by Rückbrodt & Rückbrodt (1994) were ca 600 : ca 100 : 1 (85.6% : 14.3% : 0.14%), those given by Delforge (2003) were ca 850 : 73 : 1 (92.0% : 7.9% : 0.11%), and those presented by Tyteca & Gathoye (2012) were 659 : 177 : 2 (78.6% : 21.1% : 0.24%). The corresponding numbers measured for our morphometric data-set as a result of our three summers of surveying were 141 : 53 : 2 (71.9% : 27.0% : 1.02%). In each of the three surveys, hybrids were found at only one locality: Mistéerio, Pico in the case of Delforge (2003), Toledo, São Jorge in the case of Tyteca & Gathoye (2012), and Trilho Topo, São Jorge during the present study (plants discovered by Moura in 2009). Our estimate of the percentage of hybrids was probably artificially inflated because we made particular effort in the field to locate P. micrantha in order to obtain sufficient research material. Similarly, our preference for mixed populations of P. pollostantha and P. micrantha rather than single-species populations – driven by our desire to explore the possibility of sympatric speciation – is likely to have somewhat exaggerated our estimate of the frequency of primary hybrids.

In terms of absolute numbers of plants of each species, we experienced little difficulty – at least, on the less disturbed islands – in finding populations of P. pollostantha, leaving us unable to explain the IUCN Red List statement that “there are less than five locations” (Rankou, Fay & Bilz, 2011a). Although many populations of P. pollostantha were small, a minority proved to be of considerable size. Thus, Schäfer’s (2005) estimate that 50,000–70,000 plants of P. pollostantha occur across all Azorean islands appears realistic. We were initially sceptical of his surprisingly low estimate of a total of 500–1000 flowering plants for P. micrantha. However, we recorded only 92 flowering plants at 15 localities on five islands; corresponding figures were ca 100 flowering plants at 18 localities on four islands for Rückbrodt & Rückbrodt (1994), 73 flowering plants at seven localities on two islands for Delforge (2003), and 177 flowering plants at 11 localities on two islands for Tyteca & Gathoye (2012). Thus, although Platanthera populations are fairly frequent in suitable habitats and occur on most islands, aggregating the results of these four surveys yields a remarkably low average of only 8.7 flowering plants per population of P. micrantha, and none of our study populations exceeded 20 flowering plants. Extrapolating from these figures, our estimate of total numbers of plants of P. micrantha approximates the uppermost end of the range suggested by Schäfer (2005). Lastly, our 2012 survey of the metapopulation of P. azorica on São Jorge revealed only ca 250 flowering plants; however, as the survey was conducted in dense fog and driving rain, the actual number could be larger. Thus, we calculate approximate ratios of 1:4:240 for the relative frequencies of flowering plants of P. azorica, P. micrantha and P. pollostantha, respectively. Admittedly, within each species, there are likely to exist perhaps ten non-flowering plants for every flowering plant.

Ongoing threats to the populations

Almost a century ago, Schlechter (1920) closed his account of Azorean Platantheras by warning of the threat posed to the archipelago’s vegetation by agricultural expansion. Since then, loss of the crucial mid-altitude laurisilva vegetation has been further accelerated by European Union grants. These were offered in the mid-20th century to create rough pastures for the dairy cattle that underpin the milk and cheese production, and became critical to the islands’ economy (e.g., Frey, 1977; Delforge, 2003). An assessment of species conservation priorities within the archipelago (Silva et al., 2009) that was based on a five-star categorisation awarded P. micrantha (as P. azorica) four stars and P. pollostantha (as P. micrantha) three stars, citing as serious threats to these species “Habitat degradation, expansion of agricultural land, forestation, changes in land use, invasion by alien species, herbivory, collecting, trampling, touristic use of habitat, disturbance of sensitive areas. Storms and gales, landslides. Low population density, isolation of populations, reduced area of habitat”. The subsequent IUCN Red List treatment for Azorean Platantheras (Rankou, Fay & Bilz, 2011a) conflated the two species and listed the main threats as “the destruction of the habitat to create pastures, road construction, invasive plants, tourism and plant collection”. In fact, the impacts of road construction and tourism have mostly occurred at altitudes lower than those presently favoured by the Platanthera species, and their relatively uncharismatic appearance is unlikely to cause these orchids to become high-priority targets for botanical collectors.

The main threat other than agricultural development is actually invasive plant species, which are a particular problem on this humid and largely frost-free archipelago (Sjögren, 1973; Delforge, 2003; Heleno, 2008; Silva, Ojeda Land & Rodriguez Luendo, 2009). The greatest damage is done by the Himalayan ginger, Hedychium gardnerianum, which shrouds even the steepest slopes with its rapidly growing surficial rhizomes (Fig. 3E). Invasive shrubs and trees such as the eastern Australian native Pittosporum undulatum and the Japanese native Cryptomeria japonica both eventually shade out the Platantheras, though P. pollostantha at least can persist for considerable periods of time under such light-restricted conditions; it appears to be more tolerant of habitat degradation than P. micrantha.

Formal conservation status

Cardoso et al. (2008) listed P. micrantha as one of the 37 vascular plants that number among the 100 plant and animal species most in need of conservation in the Azores. Corvelo (2010) applied IUCN conservation criteria to the 72 species and subspecies of vascular plants that were listed as endemic to the Azorean archipelago by Schäfer (2005), tentatively categorising them as 1 Extinct, 7 Critically Endangered, 20 Endangered, 18 Vulnerable, 17 Near-Threatened, 4 Least Concern, and 5 Data Deficient. The number of species labelled as Data Deficient is arguably a serious under-estimate – in our opinion, the majority of Azorean plant groups currently lack sufficient scientific knowledge to make confident conservation recommendations (cf. Schäfer et al., 2011). Having established this codicil, we note that Corvelo (2010) listed both P. pollostantha and P. micrantha as Endangered, though this decision was made on inadequate data – the estimates of plant numbers and geographical distributions given were seriously in error.

We view as particularly unfortunate the decision by the authors of the IUCN Red List (Rankou, Fay & Bilz, 2011a) to recognise only one species of Platanthera – a mythical composite species presumably largely corresponding with our P. pollostantha – as occurring in the Azores. We are also surprised that this aggregate ‘species’ was considered to merit the status of Endangered, given that in our judgement, P. pollostantha at least is not under any immediate threat. Taking into account the number and size of their populations and vulnerability of their habitats, we would categorise P. pollostantha as Vulnerable, P. micrantha as Endangered and P. azorica as Critically Endangered. We further note that P. micrantha appears to be a particularly good indicator of relatively undisturbed laurisilva habitat, and therefore constitutes a potentially valuable species for future inclusion in the EU Habitats Directive. We advocate continued mapping of populations of P. micrantha while maintaining a keen vigil for any further populations of P. azorica.

Is Platanthera azorica Europe’s rarest orchid?

The only serious competitors with Platanthera azorica for the dubious honour of being Europe’s rarest bona fide orchid species also occur in Macaronesian archipelagos, and both flower only intermittently. The first comparator, Himantoglossum metlesicsianum (formerly Barlia metlesicsiana), occurs on the Canarian island of Tenerife (Delforge, 1999; Bateman et al., 2003; Bateman & Devey, 2006; Sramkó et al., 2013). Only three populations remain on the island (cf. Stierli-Schneider, 2004; Rankou, Fay & Bilz, 2011b; Kropf, Sommerkamp & Bernhardt, 2012); one is very small and all produce few flowering plants in most years. Also occurring at high altitudes (800–1150 m asl) and in thin soils developed on neutral to slightly acidic lavas, H. metlesicsianum must contend with drier, more open habitats than those characteristic of P. azorica – indeed, in recent years it has suffered catastrophically from the effects of wildfires (Kropf, Sommerkamp & Bernhardt, 2012). Surprisingly, H. metlesicsianum is currently rated by the IUCN as Endangered rather than Critically Endangered. The second competing rarity, Goodyera macrophylla, is confined to the northern part of the island of Madeira. Seven populations reputedly totalling ca 1500 plants (albeit most of them juvenile) occur in damp acidic soils in vertiginous laurisilva cloud forests at 1000–1400 m (Delforge, 2006). This species was recently judged Critically Endangered by the IUCN and is covered by Annex II of the EU Habitats Directive (Rankou et al., 2011).

It is clear to us that the international conservation status awarded to each of the 15 Macaronesian orchids is in urgent need of revision based on solid scientific data. In the case of the rarest species, we strongly suspect that both P. azorica and H. metlesicsianum merit revision to Critically Endangered status in order to stand alongside G. macrophylla.

We thank Luis Silva, José Martins and Orlanda Moreira for assistance in the field, and the Azorean Regional Government for permission to collect research material. We also thank Henrik Pedersen and Daniel Tyteca for their well-informed and constructive reviews of the manuscript. We are grateful to Cornelia Dilger for loan of the three crucial type specimens from the Tübingen herbarium, not least for voluntarily mailing us the crucial ‘third specimen’, and to John McNeill and Leslie Lewis for their informed and detailed advice on nomenclature. We are grateful to Rob Poot for permission to use his rare cloud-free images of the ‘spine’ of São Jorge (Fig. 3A) and of flowering plants of P. algeriensis at Ifrane, Morocco (Figs. 8A–8E). Small field grants were kindly provided by the Systematics Association and the Botanical Research Fund (to RB) and the Bentham-Moxon Trust (to PR).

This paper is dedicated to Martin Engelhardt (Stuttgart), for his wisdom in bringing together Hochstetter’s holotypes in 1993–1995, his recognition of their likely taxonomic significance, and his unselfish decision not to publish that tentative insight without adequate supporting evidence.

Appendix 1 Details of Morphometric Characters Measured for Azorean Platanthera Species and their Continental Relatives (Modified from Appendix 3 of Bateman, James & Rudall, 2012)

Categories F and G were measured in the field, categories A–E in the laboratory from excised flowers (category D under a binocular microscope, category E characters 22 and 23 under a compound microscope). The character list largely follows that of Bateman, James & Rudall (2012), but includes one additional state (0) in each of characters 3 and 11, and an additional character (14A) that separates most Continental taxa from all Azorean taxa. Recording units for metric characters are given in parentheses.

Labellum (5 characters)

1. Length (0.1 mm)

2. Maximum width (excluding basal teeth, if present) (0.1 mm)

3. Reflexion, on a scale 0–4 (0 = strongly decurved, 1 = slightly decurved, 2 = vertical, 3 = slightly recurved, 4 = strongly recurved)

4. Depth of green pigmentation, on a scale 0–2 (0 = white [state not recorded among sampled individuals], 1 = pale green, 2 = dark green)

5. Maximum extent of green pigmentation (% of distance from apex to base)

Spur/ovary (5 characters)

6. Spur length (0.1 mm)

7. Spur width/mouth (0.1 mm)

8. Spur width halfway (0.1 mm)

9. Spur curvature, on a scale 1–5 (1 = strongly recurved, 2 = slightly recurved, 3 = more-or-less straight, 4 = slightly decurved, 5 = strongly decurved)

10. Ovary length (mm)

Sepals and lateral petals (5 characters)

11. Lateral sepal position, on a scale 0–3 (0 = near-vertical, 1 = substantially below horizontal, 2 = more-or-less horizontal, 3 = substantially above horizontal)

12. Lateral sepal length (0.1 mm)

13. Lateral sepal width (0.1 mm)

14. Lateral petal length (0.1 mm)

14A. Lateral petal green pigmentation (0 = near-absent, 1 = uniformly present)

Gynostemium (7 characters)

15. Maximum length of column (0.1 mm)

16. Maximum width of column (0.1 mm)

17. Maximum width of stigma (0.1 mm)

18. Length of pollinarium (0.1 mm)

19. Separation of viscidia (0.1 mm)

20. Separation of pollinia apices (0.1 mm)

21. Length of auricles (0.1 mm)

Bracts (5 characters)

22. Mean marginal cell diameter (µm)

23. Mean marginal cell shape, on a scale 1–3 (1 = barrel-shaped, 2 = subangular, 3 = angular)

24. Width floral bracts (0.1 mm)

25. Length floral bracts (mm)

26. Length basal bracts (mm)

Stem and inflorescence (4 characters)

27. Stem height, above ground level (including inflorescence) (cm)

28. Inflorescence length (mm)

29. Number of flowers/buds

30. Stem diameter (0.1 mm)

Leaves (7 characters)

31. Number of bracteoidal (cauline) leaves

32. Number of expanded (basal) leaves

33. Maximum width of longest leaf (mm)

34. Length of longest leaf (mm)

35. Position of maximum width relative to maximum length, as measured from the point of attachment to the stem (mm) [this character was used to calculate the relative position of maximum width, a proxy for leaf shape, using the formula C35 × 100/C34]

36. Degree of ‘petiole’ development, on a scale 0–2 (0 = no basal contraction, leaf lanceolate, 1 = obscure basal contraction, leaf obtuse, 2 = clear basal contraction, leaf obovate)

37. Angle of expanded leaf relative to soil surface, on a scale 1–3 (1 = 0–30°, 2 = 31–60°, 3 = 61–90°)

Additional Information and Declarations

Competing Interests

Author Contributions

Field Study Permissions

New Species Registration

Richard M. Bateman is an Academic Editor for PeerJ.

Richard M. Bateman conceived and designed the experiments, performed the experiments, analyzed the data, contributed reagents/materials/analysis tools, wrote the paper.

Paula J. Rudall and Mónica Moura performed the experiments, contributed reagents/materials/analysis tools.

The following information was supplied relating to ethical approvals (i.e., approving body and any reference numbers):

Azores Regional Government.

The following information was supplied regarding the registration of a newly described species:

The LSID for Platanthera pollostantha is 77134154-1.

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
