# Peer review of "Systematic revision of Platanthera in the Azorean archipelago: not one but three species, including arguably Europe’s rarest orchid"

_PeerJ, doi:10.7717/peerj.218_

## Round 0.1 · original submission · Minor Revisions

This is a very interesting paper. Both reviewers and I recommend minor corrections, which are mainly a few typos and clarifying, and should be easy to implement.

·

Basic reporting

No comments.

Experimental design

No comments.

Validity of the findings

No comments.

Additional comments

I would like to suggest a few minor emendations - and draw your attention to a few details that you might wish to check:

(1) The previous mismatch of names and taxa in the studied species complex is confusing, but I consider your interpretation and solution convincing. Nevertheless, I find the summarizing text (lines 94-98) slightly unclear. I recommend the following re-phrasing: "Thus, it is essential to understand that, throughout the remainder of this text, the widespread taxon long misidentified as P. micrantha is named P. pollostantha, and that the less widespread taxon long misidentified as P. azorica is named P. micrantha, whereas the true P. azorica is exceptionally rare and previously neglected. All three species are undoubtedly endemic to the Azores."

(2) I believe that, in line 399, your citation of Fig. 14C should be corrected to Fig. 14B?

(3) I believe that, in line 407, your citation of Figs. 14D and 14E should be corrected to Figs. 14C and 14D?

(4) Your crediting of "Schlechter (1992)" for making the gynostemium features more explicit (lines 532-533) is somewhat misleading, as Ruolf Schlechter himself died in 1925! I recommend that you modify the citation/reference in a way that enables you to credit the modern author who actually wrote the text concerned.

(5) Your statement in parentheses (lines 534-536) that overall labellum morphology distinguishes between Platanthera and Habenaria in the Old World is not entirely correct. Thus, the labellum is entire in a few Old World species of Habenaria (for example, among the 45 species recognized in Thailand, this is true for H. anomaliflora, H. malintana and H. mandersii). Consequently, I recommend that you delete your statement or modify it into being less categorical.

(6) As it is commonly agreed that no bursicles are formed in Platanthera, I find your reference to a "bursicular slit" (line 713) inappropriate. I suggest that you find a better term.

(7) Your reference to "Figs. 11-17" as showing scanning electron micrographs (line 1096) should be changed to "Figs 12-17" (as Fig. 11 is in reality composed of light micrographs).

(8) I recommend that, in line 1119, "fully" should be modified to "largely" or "almost fully" (as Fig. 22A does reveal a slight overlap between P. micrantha and P. pollostantha).

(9) Figs. 12D and 13D both show the median sepal. In lines 1151-1152, the citation of these figures should therefore be placed just after "sepals" (line 1151) rather than after "petals" (line 1152).

(10) In the formal descriptions, variation in metric and meristic characters is given to two standard deviations, which I find quite sensible. However, I would personally consider it more user-friendly to present the variation as ranges (as customary in taxonomic works) rather than as the mean value +/- SD.

(11) In line 1206, the statement "Designated here" is superfluous and should be deleted, as a holotype is – per definition – always designated in the protologue.

(12) Your paper is almost overwhelmingly detailed, which I basically find very useful. In just a few cases, however, I think that the text has become excessively long-winded. This is particularly true for the nomenclatural discussion under "The (regrettable) solution" (lines 1381-1452), especially your extensive quoting of articles from the nomenclature code. Obviously, a nomenclatural discussion is needed, but I recommend that you reconsider the length and wording of this section.

(13) In your (relevant!) discussion of shrinkage of organs in connection with the preparation of herbarium specimens, I would personally be reluctant to try and quantify the shrinkage based on comparisons between type specimens and average values from modern population samples (lines 1489-1493). After all, you cannot be certain that the type specimens were not relatively small representatives of their species already at the time of collection.

Apart from these minor points, I consider your manuscript an excellent piece of work.

·

Basic reporting

No comment

Experimental design

No comment

Validity of the findings

No comment

Additional comments

Basic Reporting

I found the paper both scientifically sound and enthralling, reporting on an important issue that required clarification, especially in view of the new, original (re-) discovery of the “third Azorean Platanthera species”. The results and conclusions from the research are of uppermost interest for both the community of orchidologists and the authorities responsible for nature conservation. The paper is rather long, but in order to be comprehensive and understandable, such long developments were unavoidable.

Experimental Design

The methods applied are quite appropriate. The paper concentrates on morphological aspects, morphometrics, taxonomy, systematics, and nature conservation aspects, leaving to a companion paper the genetic and molecular aspects, as well as relationships with mycorrhizae. There is room for further research, especially (as recognised by the authors) as regards the pollinator attraction strategies, identification of pollinators, and characterisation and exploitation of fragrances developed by all three Azorean Platanthera species. However, as such, the use of the methods listed above are more than sufficient to give a convincing view of the systematics of Platanthera in the Azores and their relationships with species complexes from the Old World.

Validity of the Findings

See above for some of my comments regarding results. In addition, please consider the following. In the summary, the option you take is that of only one immigration event: “an initial anagenetic speciation event, aided by the founder effect, was followed by the independent origins of at least one of the two rarer endemic species from within the first-formed endemic species, through a cladogenetic speciation process that involved radical shifts in floral development, considerable phenotypic convergence [sic: see corrections below], and increased mycorrhizal specificity” (the “at least one” maintains some ambiguity). However, the question is more open in the text: for example, see the divergent conclusions drawn from a morphological approach (two immigration events, one giving rise to P. azorica, derived from a P. chlorantha-like ancestor, and one giving rise to the other two Azorean species, derived from P. bifolia) and a molecular approach (one immigration event, in which a reversion in characters gave rise to P. azorica, after a first anagenetic speciation event gave rise to one of the other Azorean species, presumably P. pollostantha). See also the caption of Figure 25, with spider diagrams comparing characters of “(A) Platanthera bifolia and its putative Azorean descendants, P. micrantha and P. pollostantha”, and “(B) Platanthera chlorantha, P. algeriensis, P. homboei and their putative Azorean descendant, P. azorica”. The ambiguity between the two scenarios appears in other parts of the text. In my opinion, you should state more explicitly, either that both scenarios are equally probable, and that it is presently impossible to decide which one is the most likely, or that one of the two scenarios has to be favoured (and for which reasons). This should appear both in a paragraph in the discussion, and in the summary.

Other comments

1. You considered that the mainland European populations of Platanthera are represented by only four species, namely P. chlorantha, P. bifolia, P. algeriensis and P. holmboei (e.g., in the summary, you stress “all four continental European relatives in the P. bifolia-chlorantha aggregate”). But several authors maintain that more species exist (sometimes considered at the sub-specific or varietal level), essentially P. fornicata (BUTTLER 2011) and P. lesbiaca (DEVILLERS et al. 2010, 2012). Most probably you will not acknowledge these “species”, but you might consider including a word of discussion, on the variability of Platanthera species and the various concepts of different authors.

2. Lines 865 – 867: It may be relevant to mention that we (Tyteca & Gathoye 2012) observed “enormous” individuals of P. pollostantha in São Jorge, one bearing no less than 163 flowers (only 44 of them open!), and reaching 61 cm in height, with an inflorescence length of 23 cm!

3. Lines 985 – 988: “a relatively high percentage of ocean island endemic species become autogamous, presumably to free them from reliance on what will at best be a limited spectrum of potential pollinators”. A reference would be worth giving here.

4. Paragraph in lines 994 – 1005: Two pictures published by us (Tyteca & Gathoye 2012) show pollinia distributed among various parts of inflorescences of P. pollostantha (including, e.g., a bract). This might be used an indirect proof of insect visitation, because otherwise, how could these pollinia be found in such situations? One of the pictures also shows two small flies in the inflorescence, but it is unlikely that these play any role in pollination.

Various corrections

Line 41: divergence and not convergence

Line 177: constituted and not consitituted

In Figure 1: It is very hard to distinguish red and orange colours.

Line 2031: Caldeira and not Caldera (also probably in the text)

Line 379: distinguish and not distingush

Line 432: four and not three

Line 582: relative and not relatve

Lines 2008 – 2010: the exact reference for Tyteca & Gathoye is
Tyteca D, Gathoye J-L. 2012. Orchidées des Açores: compte rendu d’un voyage à Pico et São Jorge, du 11 au 21 juin 2012. Jornal da Associação de Orquídeas Silvestres – Portugal n° 2 : 26-43.

---

## Round 0.2 · accepted · Accept

I'm happy with the corrections have been made.